

# Gross and net land cover changes based on plant functional types derived from the annual ESA CCI land cover maps

Wei Li[1], Natasha MacBean[2], Philippe Ciais[1], Pierre Defourny[3], Céline Lamarche[3], Sophie Bontemps[3], Richard A. Houghton[4], Shushi Peng[5]

[1]Laboratoire des Sciences du Climat et de l'Environnement, LSCE/IPSL, CEA-CNRS-UVSQ, Université Paris-Saclay, 91191 Gif-sur-Yvette, France

[2]School of Natural Resources and the Environment, The University of Arizona, Tucson, AZ, 85721, USA

[3]Earth and Life Institute-Environmental Sciences, Université catholique de Louvain, Louvain-la-Neuve, Belgium

[4]Woods Hole Research Center, Falmouth, Massachusetts, USA

[5]Sino-French Institute for Earth System Science, College of Urban and Environmental Sciences, Peking University, Beijing 100871, China

*Correspondence to:* Wei Li (wei.li@lsce.ipsl.fr)

**Abstract.** Land-use and land-cover change (LULCC) impacts local energy and water balance and contributes at global scale to a net carbon emission to the atmosphere. The newly released annual ESA CCI land cover maps provide continuous land cover changes at 300 m resolution from 1992 to 2015, and can be used in land surface models (LSMs) to simulate LULCC effects on carbon stocks and on surface energy budgets. Here we investigate the absolute areas, gross and net changes of different plant functional types (PFTs) derived from ESA CCI products. The results are compared with other datasets. Global areas of forest, cropland and grassland PFTs from ESA are 30.4, 19.3 and 35.7 million $km^2$ in 2000. The global forest area is lower than that from LUH2v2h (Hurtt et al., 2011), Hansen et al. (2013) and Houghton and Nassikas (2017) while cropland area is higher than LUH2v2h (Hurtt et al., 2011), in which cropland area is from HYDE3.2 (Klein Goldewijk et al., 2016). Gross forest loss and gain during 1992-2015 are 1.5 and 0.9 million $km^2$ respectively, resulting in a net forest loss of 0.6 million $km^2$, mainly occurring in South and Central America. The magnitudes of gross changes of forest, cropland and grassland PFTs in ESA CCI are smaller than those in other datasets. The magnitude of global net cropland gain for the whole period is consistent with HYDE3.2 (Klein Goldewijk et al., 2016), but most of the increases happened before 2004 in ESA while after 2007 in HYDE3.2. Brazil, Bolivia and Indonesia are the countries with the largest net forest loss from 1992 to 2015, and the decreased areas are generally consistent with those from Hansen et al. (2013) based on Landsat 30 m resolution images. Despite discrepancies compared to other datasets, and uncertainties in converting into PFTs, the new ESA CCI products provide the first detailed long time-series of land-cover change and can be implemented in LSMs to characterize recent carbon dynamics, and in climate models to simulate land-cover change feedbacks on climate. The annual ESA CCI land cover products can be downloaded from http://maps.elie.ucl.ac.be/CCI/viewer/download.php (Land Cover Maps – v2.0.7; see details in **Section 2.5**).



## 1 Introduction

Land-use and land-cover change (LULCC) is the essential human perturbation on natural ecosystems (Klein Goldewijk et al., 2016) and one of the main drivers of climate change (Alkama and Cescatti, 2016; Bonan, 2008) through biophysical (e.g. albedo and transpiration change) (Peng et al., 2014; Zhao and Jackson, 2014) and biogeochemical effects (e.g. carbon

emissions from gross deforestation and carbon sinks in secondary forest regrowth) (Houghton and Nassikas, 2017). Forest loss from 2003 to 2012 was found to have caused a local increase in air temperature of about 1 °C in temperate and tropical regions, despite less solar energy being absorbed by non-forest secondary vegetation with a higher albedo (Alkama and Cescatti, 2016). Global net LULCC carbon emissions ($E_{LUC}$) are estimated to be 1.1 ±0.4 Pg C yr$^{-1}$ during the past decade (2006-2015) by the bookkeeping model of Houghton and Nassikas (2017) based on the national land cover data from Food

and Agriculture Organization (FAO). The $E_{LUC}$ diagnosed from an ensemble of land surface models (LSMs) is 1.3 ±0.3 Pg C yr$^{-1}$ during 2006-2015 (Le Quere et al., 2016) based on different (successive) versions of expanding cropland and pasture area from the HYDE dataset (Klein Goldewijk et al., 2016).

Accurate, well defined, and spatially explicit gridded LULCC data are a prerequisite for calculating $E_{LUC}$ in models, either under the form of annual area change in bookkeeping models or converted to changes in plant functional type (PFT) areas in

LSMs. In fact, uncertain historical LULCC data are one of the largest contributors to the uncertainties in $E_{LUC}$ estimation (Bayer et al., 2017; Houghton and Nassikas, 2017). In addition to the inventory data (e.g. FAO data reported by individual countries), satellite observations in the recent three decades offer the possibility to characterize the vegetation distributions as well as their temporal changes due to both natural and anthropogenic activity. Global satellite data include the Global Land Cover 2000 (GLC2000) map based on SPOT VEGETATION (SPOT-VGT) (1 km resolution) (Bartholomé and

Belward, 2005), the MODIS Collection 5 Land Cover Product (500 m resolution) (Friedl et al., 2010), forest cover maps based on Landsat (30 m resolution) (Hansen et al., 2013), the GlobCover 2005 and 2009 products (300 m resolution) (Bontemps et al., 2011; Defourny et al., 2012) and European Space Agency Climate Change Initiative (ESA CCI) epoch maps based on MERIS (300 m resolution) (Bontemps et al., 2013). These satellite land cover products, however, differ in terms of land cover type, spatial resolution, time span, stability and accuracy due to the different sensor designs,

classification procedures and validation methods (Bontemps et al., 2012). In order to use satellite land cover (LC) products in LSMs, these maps of LC classes are usually translated into maps of PFTs to drive the carbon dynamics in vegetation and soils (Poulter et al., 2015); however, the cross-walking table between LC classes and PFTs is complicated by subjective decisions related to the interpretation of LC class descriptions, and therefore is a source of uncertainty in model simulations (Hartley et al., 2017). Because LC transitions of opposite directions can happen simultaneously in a 0.5° × 0.5° grid cell,

which is a typical spatial resolution of LSMs, gross transitions instead of net transitions are gradually implemented in LSMs to more accurately simulate $E_{LUC}$ (Bayer et al., 2017; Shevliakova et al., 2009; Stocker et al., 2014; Wilkenskjeld et al., 2014; Yue et al., 2017). Thus, high-resolution and successive long-term data on LC change are needed to generate the gross transition matrix used in LSMs. Although the products from Hansen et al. (2013) have a high resolution (30 m), they only



provide forest area change rather than changes between all LC types. Further, the gross forest gain is only available for the whole period of 2000-2012 rather than at annual time step (Hansen et al., 2013). The previous ESA CCI epoch maps contain all LC types (Bontemps et al., 2013) but the LC transitions are not appropriate to be used in LSMs because these epoch products represent five-year composite maps and thus do not allow to assess annual LC change dynamics, and furthermore

only transitions to or from forest cover were considered at that time (Li et al., 2016).

The newly released annual ESA CCI land cover maps from 1992 to 2015 partly overcome these challenges with 300 m resolution and long and successive annual time series for all major land cover transitions (i.e. the maps now include transitions between non-forest classes, including grasses, crops and urban areas) (ESA, 2017) and thus can be potentially translated into PFT maps used in the LSMs. The objectives of this study are to document the major gross and net changes

and transitions in PFT maps derived from annual ESA CCI LC products and to evaluate whether they can be used in LSMs. Geographical distributions and temporal trends of the translated PFT maps from ESA CCI products are characterized and compared with those from other datasets. It should be noted that our analyses are based on the PFT maps that have been translated from the ESA CCI LC maps, rather than the original LC classes, because we aim to demonstrate the differences between different datasets and provide suggestions to modellers for implementing them in LSMs.

## 2 Methods

### 2.1 ESA CCI land cover products

The annual ESA CCI LC maps cover a period of 24 years from 1992 to 2015 at a spatial resolution of 300 m (ESA, 2017). These maps describe the Earth terrestrial surface in 37 original LC classes based on the United Nations Land Cover Classification System (UN-LCCS) (Di Gregorio, 2005).

This unique long-term land cover time series was achieved by combining the global daily surface reflectance of 5 different observation systems while aiming to maintain a good consistency over time. This was identified as a key requirement from the modeling community (Bontemps et al., 2012). Each of these global daily measurements of multispectral radiance recorded from 1992 to 2015 have been pre-processed to complete radiometric calibration, and geometric and atmospheric correction, as well as clouds and clouds shadows screening. The full archive of MERIS (2003-2012) providing 15 spectral

bands at 300m resolution was classified to establish a baseline by fusing the outputs of machine learning and unsupervised algorithms (ESA, 2017). The 1 km time series recorded respectively by AVHRR from 1992 to 1999, SPOT-VGT from 1999 to 2013, and PROBA-V from 2014 and 2015 were used to detect and confirm the change which was eventually delineated more precisely at the 300m spatial resolution whenever possible, i.e. later than 2004. This last step results in both back- and forward-dating the 10-year baseline LC map to produce the 24 annual LC maps from 1992 to 2015. In order to avoid false

change detections due to the inter-annual variability in classifications, each a change has to persist over more than two successive years in the classification time series to be confirmed (for more information see Section 3.1.2 of the ESA CCI LC Product User Guide, ESA, 2017)). The resulting series of consistent 300m annual LC maps from 1992 to 2015 is delivered





with a pixel-based uncertainty value indicating the confidence at which a LC class was assigned for each pixel. The accuracy of ESA CCI LC products was evaluated at global scale. An object-based validation database of 2600 Primary Sampling Units was built by a panel of international experts to specifically assess the accuracy of both the LC classes and change (ESA, 2017).

## 2.2 PFT area and net change

The original 37 ESA CCI LC classes were first aggregated into $0.5° \times 0.5°$ resolution and then translated into 14 different PFTs based on the cross-walking table (Table S1) from the ESA Land Cover Product User Guide (ESA, 2017). This table originated from Poulter et al. (2015) and was further adjusted for some classes due to improved understanding of how the LC class descriptions can be interpreted to estimate fractional cover of PFTs from each LC class, in particular for mosaic classes and sparsely vegetated regions. PFTs were grouped into major vegetation types: forest, shrub, grassland and cropland. The tree PFTs and shrub PFTs (Table S1) were summed to obtain the forest and shrub area respectively; thus, the shrub PFTs are excluded from tree PFTs in our analyses. The net area change was calculated by comparing two annual PFT maps at $0.5° \times 0.5°$ resolution.

## 2.3 Gross PFT changes and transitions

Gross changes need to be considered differently because it is only possible to derive the net change by comparing the annual maps sequentially. Gross changes may be far larger than the net changes, and thus may show different magnitudes or even directions of LULCC fluxes when simulated in LSMs. To document all the bidirectional LC transitions at $0.5° \times 0.5°$ resolution, high-resolution LC transitions data are needed. Therefore, the annual ESA CCI LC maps are compared year by year at 300 m resolution to record the gross loss and gain of each original LC class over the whole period from 1992 to 2015. There are 23 original LC classes that experienced gross changes (classes with stars in Table S1).

In order to derive the gross transitions, all possible transitions (506 in total) between the 23 original LC classes with gross changes were calculated at 300m resolution. There are a total of 422 gross transitions between these 23 original LC classes. These gross changes in the original classes were then translated into gross changes of PFTs using the LC-to-PFT cross-walking table (Table S1) and grouped into the major vegetation types (forest, shrub, grassland, cropland). For example, a LC transition from class "50", corresponding to 90% tree PFT in Table S1, to class "30" (10% tree PFT) is taken as a forest loss of 80% in that 300 m grid cell. Finally, the converted transitions were aggregated into fractions in each $0.5° \times 0.5°$ grid cell.

## 2.4 Comparison with other datasets

Three land-use and land-cover datasets (Table 1) were used for comparison, namely, forest, grassland and cropland area from Land Use Harmonization (LUH2v2h) data (Hurtt et al., 2011), forest cover data from Hansen et al. (2013) and national forest area data from Houghton and Nassikas (2017). The cropland and pasture areas in LUH2v2h dataset are from HYDE3.2 (Klein Goldewijk et al., 2016), in which ESA CCI epoch LC map in 2010 (representing 2008-2012) was used as a



spatial reference map for the area allocation and the national cropland and grazing land were adjusted to match the FAO STAT data (FAOSTAT, 2015) as close as possible. The national forest areas from Houghton and Nassikas (2017) are based on FAO Forest Resources Assessment (FRA) data. Thus, these two additional sources of data, HYDE3.2 (Klein Goldewijk et al., 2016) and FAO FRA (FAO, 2015), were not shown in the figures.

It should be noted that land use data are not necessarily the same as land cover, and the exact definitions and categorization of forest (cropland and grassland) are different for each dataset (see details in **Discussion**). Nevertheless, these represent the best datasets available for comparison, and we have tried to harmonize the definitions where possible (see below), but to some degree this is an ongoing discussion between the modeling and data communities. Furthermore, all the LSMs have to use these datasets for deriving PFT changes back through time, so it is a very worthwhile exercise to determine if the broad
groupings differ, and to what extent.

Absolute areas, net changes and gross transitions from 1992 to 2015 in the LUH2v2h dataset (Hurtt et al., 2011) were used for comparison. Forest used in this study from LUH2v2h (Hurtt et al., 2011) refers to the total of primary and secondary forest; cropland refers to all crop types; grassland refers to the total of pasture and rangeland. Because LUH2v2h data use cropland and grazing land areas from HYDE3.2 as an input (Hurtt et al., 2011), the spatial distributions are mainly
determined by HYDE3.2. The gross transitions in LUH2v2h data are calculated from the Global Land use Model (Hurtt et al., 2006) that tracks sub-grid cell loss and gain in land use categories. They first determined the urban area in each grid cell proportionally from cropland, pasture and secondary lands, and if these areas cannot fulfill the urban increase, primary lands were cleared. The minimum transition rates between cropland, pasture and other (sum of primary and secondary lands) were then calculated to identify the gross transitions between these land use categories (Hurtt et al., 2011). Transitions related to
shifting cultivation and wood harvest were determined last (Hurtt et al., 2011).

Only annual gross forest loss each year during 2000-2014 and total gross forest gain during 2000-2012 are available in the dataset of Hansen et al. (2013). Thus, the net forest area change from this dataset only refers to the period of 2000-2012. The national forest area data from 1992 to 2015 in the dataset of Houghton and Nassikas (2017) were used to calculate the forest area changes.

A land mask with nine regions (Figure 1) defined by Houghton (1999) was used to derive the regional values.

**2.5 Data availability**

The ESA CCI LC maps can be viewed online using http://maps.elie.ucl.ac.be/CCI/viewer/index.ph, and the data products can be download from http://maps.elie.ucl.ac.be/CCI/viewer/download.php. After entering some basic information, the land cover maps with a specific version number are available for download in the Climate Research Data Package (CRDP)
section. In this study, we used the version: "Land Cover Maps – v2.0.7". A protocol of translating the original ESA CCI LC maps into PFT maps and an example of LC map and PFT map in 2000 used in this study can be downloaded from doi: https://doi.org/10.5281/zenodo.834229



## 3 Results

### 3.1 PFT areas in year 2000

After translating the original ESA CCI LC classes into PFTs using the cross-walking table (Table S1), the global and regional areas of forest, cropland and grassland PFTs in year 2000 are shown in Figure 1. Global areas of forest, cropland

and grassland PFTs are 30.4, 19.2 and 35.7 million $km^2$, respectively. Global forest area is 6.7, 1.8 and 10.1 million $km^2$ lower than that from LUH2v2h (Hurtt et al., 2011), Hansen et al. (2013) and Houghton and Nassikas (2017), respectively. It is also much lower than the recently reported global forest area of 43.3 million $km^2$ with increased forest area estimate in dryland biomes using Google Earth images (Bastin et al., 2017). Global cropland area from ESA CCI is 4.2 million $km^2$ larger than that from LUH2v2h, while the difference in global grassland area is relatively small.

Forest area from ESA CCI is slightly lower than that from Hansen et al. (2013) in the regions where most of forests are distributed, i.e. South and Central America, tropical Africa, North America and the former Soviet Union. Forest area from LUH2v2h (Hurtt et al., 2011) is larger than that from ESA CCI in most regions except in South and Central America, tropical African and Pacific developed region. Forest area from Houghton and Nassikas (2017), however, is systematically higher than that from ESA CCI in all regions. Cropland area from ESA CCI matches that from LUH2v2h (Hurtt et al., 2011)

in North America but is higher in all the other regions. Although the global grassland area is similar between ESA CCI and LUH2v2h (Hurtt et al., 2011), larger differences are seen at regional scale. Grassland area from ESA CCI was found to be much higher than that from LUH2v2h (Hurtt et al., 2011) in North America and the former Soviet Union (4.0 and 3.5 million $km^2$ higher, respectively) but much lower (2.4 million $km^2$) in North Africa and Middle East.

### 3.2 Gross area change

#### 3.2.1 Time series of gross PFT change

After translating all the 422 gross transitions detected between the original ESA LC classes into PFTs, the time series of gross changes of PFTs are shown in Figure 2. Generally, the gross changes are related to the net, i.e., where there are more gross changes, more net changes can be found. Major gross changes occur in forest, cropland and grassland PFTs, with a global gross gain of 0.91, 1.2 and 1.1 and a global gross loss of 1.5, 0.56 and 0.98 million $km^2$ respectively, from 1992 to

2015. The magnitudes of gross changes of these three PFTs are larger before 2005 than after 2005. Especially during the late 1990s, both intensive gross forest loss and gain occurred but overall resulted in net forest loss. Accordingly, both gross and net cropland area expands during this period. Two other peaks of net forest loss were found in 1995 and 2004, during which net cropland area increased. Although grassland experienced large gross loss and gross gain, the net area remains stable, except in 2004 where a net increase was found.

The temporal correlations of gross and net changes between ESA CCI PFTs, Hansen et al. (2013) and LUH2v2h (Hurtt et al., 2011) are not significant (p > 0.05, Table S2). The magnitudes of gross changes of forest from LUH2v2h (Hurtt et al., 2011) and Hansen et al. (2013) and cropland from LUH2v2h (Hurtt et al., 2011) are much larger than those detected from ESA CCI



PFT maps (Figure 2). In contrast to gross changes of forest and cropland from ESA CCI maps, annual gross changes from LUH2v2h (Hurtt et al., 2011) show larger variations after 2005 than before 2005. Especially before 2000, the annual gross changes of forest and cropland from LUH2v2h (Hurtt et al., 2011) are constant because HYDE3.2 provide cropland and pasture area only at a 10 year time step before 2000 (at an annual time step after 2000), and a linear interpolation was used in

LUH2v2h (Hurtt et al., 2011) to produce the annual maps from HYDE3.2 before 2000. The net forest loss and corresponding net cropland gain in 2004 coincides in the ESA CCI PFT maps and LUH2v2h (Hurtt et al., 2011) but the years of cropland gain in HYDE3.2 are rather different from in ESA CCI PFTs during the other periods. Although the difference in the magnitude of gross grassland changes between ESA CCI PFTs and LUH2v2h (Hurtt et al., 2011) is relatively smaller than that of forest and cropland, the net grassland changes are not consistent over time.

Gross changes of shrub and bare soil are also detected over the whole period, and the net changes of these PFTs is generally a loss in area. The magnitudes of gross water body area changes are small compared to other PFTs. There is relatively large net increase during 1995-2000 and moderate net decrease during 2000-2010. Urban areas keep expanding over the whole period, and the increasing rates are high during 2001-2004 and 2012-2014.

### 3.2.2 Spatial distributions of gross PFT changes

The spatial distributions of net and cumulative gross changes of forest, cropland and grassland PFTs between 1992 and 2015 are shown in Figure 3, and the distributions of the other PFTs are shown in Figure S1. Intensive gross forest loss and sparse gross forest gain in South America result in a strong net decrease of forest area (Figure 3). There are also considerable gross and net forest loss in South and East Asia and in some regions of tropical Africa. Gross forest gain occurs pervasively in boreal regions. Some regions of intensive gross forest gain were found in South Asia, tropical Africa and South America, but

with a small extent. Gross cropland gain occurs all over the world, and especially in South America, tropical Africa (particularly in the Sahel), South and Southeast Asia and Central Asia. By contrast, gross cropland loss is only observed in Europe and across the North China Plains. The cropland loss in these two regions is mainly caused by urbanization and thus an increase of urban area was found (Figure S1). Therefore, the net cropland change is an increase in most regions except Europe and the North China Plains. Grassland in temperate and tropical regions experienced extensive gross gain and gross

loss, but the gross gain and loss are not fully coincident, leading to a pattern of coexisting net gain and loss everywhere (Figure 3). The changes in grassland are relatively small in boreal region.

The changes to shrubs are largely distributed in tropical regions, with a net gain in South America and net loss in tropical Africa and South Asia (Figure S1). Intensive gross changes of bare soil were found in North China, Central Asia, Australia and the south edge of Sahara, mainly caused by the gross transitions between original ESA LC classes "200" (bare areas)

and "150" (sparse vegetation; tree, shrub, herbaceous cover <15%). Water body changes are relatively small compared to other PFTs. In addition to the urban area increase over cropland in Europe and North China Plain, there is also urban expansion to cropland in United States (Figure S1).



### 3.3 Net area change of PFTs

#### 3.3.1 Global change

The global and regional net area changes of forest, cropland and grassland PFTs from ESA CCI LC maps since 1992 are shown in Figure 4 (solid lines). Global net forest loss and net cropland gain between 1992 and 2015 are 0.60 and 0.67

million km$^2$, respectively. Global forest area decreased fast from 1992 to 2004 accompanied by fast increases of cropland. Forest area stayed stable between 2004 and 2009 and then decreased again, although by a smaller magnitude than in 1992-2004, during the recent period from 2009 to 2015. Meanwhile, cropland area remains relatively stable since 2004. Net grassland changes are small compared to forest and cropland changes.

The magnitudes of net forest area change from LUH2v2h (Hurtt et al., 2011)) are much smaller than those from ESA CCI,

mainly because the forest area decrease between 1992 and 2009 (Figure 4) is not reflected in the LUH2v2h dataset (Hurtt et al., 2011). Although the net cropland area increases from 1992 to 2015 are similar between ESA CCI and LUH2v2h (Hurtt et al., 2011), the temporal trajectories are rather different. The increase of cropland in ESA CCI data happened between 1992 and 2004, while cropland area in LUH2v2h (Hurtt et al., 2011) mainly increased since 2007 (Figure 4). Grassland area changes in LUH2v2h (Hurtt et al., 2011) display more variations than those from ESA CCI. There was an increase in

grassland in LUH2v2h (Hurtt et al., 2011) in the earlier period (1992-2004) where ESA CCI had the increase in cropland. Globally, net forest area loss between 1992 and 2015 from both Hansen et al. (2013) and Houghton and Nassikas (2017)) is much larger than that from ESA CCI and LUH2v2h data (Hurtt et al., 2011).

#### 3.3.2 Regional change

Consistent with the spatial distributions of net forest change in Figure 3, net forest loss in South and Central America

dominates the global net forest loss (Figure 4), accounting for 75% of the global total. The magnitude of net forest loss is close to that observed by Hansen et al. (2013) in this region. However, the magnitudes of net forest loss from ESA CCI PFTs in other regions are generally smaller than those from Hansen et al. (2013). Net forest area change from Houghton and Nassikas (2017) also shows a stronger loss in all three tropical regions than that in other datasets, especially in South and Central America and tropical Africa. It should be noted that the net forest loss in South and Southeast Asia is consistent

between LUH2v2h (Hurtt et al., 2011), Hansen et al. (2013) and Houghton and Nassikas (2017), and all these datasets have much larger net forest area loss than ESA CCI data. All datasets demonstrate net forest gain in North America, except Hansen et al. (2013), which has a strong forest loss. The forest area in LUH2v2h data (Hurtt et al., 2011) and inventory-based data from Houghton and Nassikas (2017) shows a net increase in China region and western Europe. In contrast, forest area in the satellite-based datasets of ESA CCI PFTs and Hansen et al. (2013) is stable or slightly decreasing.

South and Central America, tropical Africa and the former Soviet Union are the regions with largest contributions to the global total net cropland increase, representing 37%, 33% and 11% of the global total. The regional patterns of temporal net cropland area change are rather different between ESA CCI PFTs and LUH2v2h (Hurtt et al., 2011) although the global net



changes from 1992 to 2015 are similar. Cropland from LUH2v2h (Hurtt et al., 2011) expands more in tropical regions but decreases more in other regions than in ESA CCI PFTs (Figure 4).

Grassland area from ESA CCI PFTs slightly increases in South and Central America and South and Southeast Asia, and slightly decreases in North America, the former Soviet Union and North Africa and Middle East. Differences in grassland

change are large between ESA CCI PFTs and LUH2v2h (Hurtt et al., 2011) in all regions other than tropical regions.

### 3.3.3 Countries with largest net forest area loss and gain

Countries with the largest net forest PFT area loss between 1992 and 2015 from ESA CCI maps are shown in Figure 5, and countries with the largest net forest PFT gain in Figure 6. Brazil, Bolivia and Indonesia are the three countries with largest net forest losses during 1992-2015 with a net loss of 0.28, 0.044 and 0.042 million $km^2$, respectively. The net forest loss in

Brazil during the whole period is consistent between ESA CCI PFTs, LUH2v2h (Hurtt et al., 2011) and Hansen et al. (2013), despite the fact that temporal patterns are different between ESA CCI and LUH2v2h (Hurtt et al., 2011). Net forest changes between ESA CCI PFTs and Hansen et al. (2013) are also similar in Indonesia, Argentina, and Cambodia, while net forest loss in Russia and Congo, DRC from Hansen et al. (2013) is much larger than that from ESA CCI. Net forest loss from Houghton and Nassikas (2017) is always higher than the loss from other datasets in the all these countries except in China

and Russia where a net forest gain was found in Houghton and Nassikas (2017).

The overall net cropland gain from 1992 to 2015 between ESA CCI and LUH2v2h (Hurtt et al., 2011) is similar in Bolivia but is rather different in all the other countries in Figure 5. Larger cropland gain from LUH2v2h (Hurtt et al., 2011) compared to ESA CCI was found in Brazil, Indonesia, Argentina, and Paraguay while lower cropland gain was found in Cambodia and Congo, DRC. The cropland area change in China and Russia from LUH2v2h (Hurtt et al., 2011) shows even a

net loss rather than gain. Grassland area increased in Argentina, Paraguay, Russia, Cambodia and Congo, DRC in LUH2v2h (Hurtt et al., 2011), which was not captured by ESA CCI maps.

The magnitudes of forest change in the countries with the largest forest gain in Figure 6 are much smaller than those with largest forest loss (Figure 5). For example, the net forest gain from 1992 to 2015 is 0.019 million $km^2$ in Canada, compared with a forest loss of 0.28 million $km^2$ in Brazil. In these largest forest gain countries, forest area change from Hansen et al.

(2013) indicates a net forest gain only in Uruguay, and a net loss or stable in other countries. Again, contrary to ESA CCI PFTs, Houghton and Nassikas (2017) forest area data show large magnitudes of net forest loss in Myanmar, Sudan and Nigeria, and greater magnitudes of net forest gain than other datasets in Uruguay. Cropland changes from LUH2v2h (Hurtt et al., 2011) display larger magnitudes, more variations and even different directions than those from ESA CCI in these nine countries in Figure 6. Grassland area changes from ESA CCI are rather flat, which is different from those in LUH2v2h

(Hurtt et al., 2011).

## 4 Discussion



## 4.1 Differences in total area of forest, cropland and grassland

The forest, cropland and grassland areas from different datasets do not match at global or regional scales (Figure 1), mainly caused by the differences in land cover definitions and data sources (Table 1), as well as the uncertainties in the cross-walking table used for translating original ESA CCI LC classes into PFTs. The canopy cover of forest varies in different

ESA CCI LC classes with defined ranges such as >15%, 15-40%, and >40% depending on the "openness" of the canopy and according to the UN-LCCS framework provided by the FAO (Di Gregorio, 2005). Although continuous tree cover fractions are provided in data from Hansen et al. (2013), the forest cover is defined as >25% canopy closure for trees higher than 5m (Hansen et al., 2010). Forest areas in Hansen et al. (2013) are obtained from NASA's Landsat instruments with a high spatial resolution of 30m that can capture the small-scale forest areas. This partly explains the larger forest extent in Hansen et al.

(2013) than ESA CCI PFT maps. It seems that forest area from ESA CCI PFTs is higher than that from Hansen et al. (2013) in arid regions but lower in humid regions (Figure S2).

The definition of forest by FAO, which is the data source of Houghton and Nassikas (2017), is a canopy cover >10%. FAO's forest areas are based on reports from the member countries (FAO, 2015) and the methods of compiling data in each country may vary largely, e.g. from field survey or from satellite imagery based estimation (Grainger, 2008; Harris et al., 2012).

Furthermore, in the definition of forest by FAO, natural disturbance suppressing forests are taken as a forest, but from satellite, they are not detected as forest cover.

Forest area estimates in LUH2v2h (Hurtt et al., 2011) are based on aboveground biomass density from Miami-LU ecosystem model (Hurtt et al., 2006), and cropland and pasture areas are based on HYDE 3.2 (Klein Goldewijk et al., 2016). HYDE 3.2 uses the cropland and pasture areas from FAO STAT (FAOSTAT, 2015) as the main land-use input data and the ESA CCI

epoch LC map of 2010 as a spatial reference map (Klein Goldewijk et al., 2016). Thus, the grasslands in LUH2v2h refer to the sum of intensively managed pastures and less intensively used rangelands (Klein Goldewijk et al., 2016), while the grassland PFT from ESA CCI maps also includes natural grassland, which may be the reasons for less grassland in LUH2v2h (Hurtt et al., 2011) than ESA CCI, especially in the former Soviet Union, western Europe and North America (Figure 1).

The final spatial area of each PFT in this study is derived from a combination of ESA LC map and the cross-walking table (Table S1) used for translating original ESA LC classes into PFTs. The range in tree cover canopy openness (as discussed above) and percent of each type of vegetation for mosaic LC classes in the LC description contributes to uncertainty in the conversion fractions used to translate the LC classes into PFT in the cross-walking table. Thus, uncertainties in the cross-walking table contribute to the differences in forest, cropland and grassland PFT areas when comparing with other datasets.

Only one value is used to prescribe the fraction of each PFT for a given class, e.g. class "50" corresponds to 90% of broadleaf evergreen trees in Table S1. This hinders an explicit representation of spatially heterogeneous tree cover fractions. In the absence of other information, the approximate mid-point of the range in the LC class description is used when calculating the fraction of forest PFT from a given LC class. For example, class "61" represents a closed canopy (>40%) and



therefore we use a LC to tree PFT conversion fraction of 70% (0.7) as the mid-point between 40 and 100% (Table S1). Class "62" on the other hand is an open canopy (15-40% cover) and therefore we use a LC to tree PFT conversion factor of 30%. Some exceptions to this general rule are made when we have a better understanding of the species or biomes included in a given LC class. For example, class "50" (broadleaved evergreen trees) encompasses tropical rainforests. Although the class

description states that the canopies in this class can be closed to open (>15%), we know that the tree cover fraction is much higher than a mid-point of ~60%, therefore we use a conversion factor of 90%. However, this level of knowledge is not available for all LC classes. This is particularly true for mosaic and sparse vegetation classes (e.g. classes "100", "110" and "150") that span different regions/biomes that may contain different fractional coverage of vegetation.

Likewise, an explicit regional classification is required for cropland. For example, class "10" (cropland, rainfed) is separated

well in North America, i.e., mainly partitioning into class "11" (herbaceous cover), and thus the cropland area in this region is highly consistent with LUH2v2h data (Hurtt et al., 2011) (Figure 1). In tropical Africa where class "10" is not separated into a more detailed classification, the difference in cropland areas between these two datasets are large (Figure 1). This is because if most of the cropland in this region belongs to class "12", using the corresponding value for class "10" in the cross-walking table (90% for class "10" vs. 30% for class "12", Table S1) overestimates cropland areas.

Hartley et al. (2017) also investigated the uncertainty in simulations of carbon, water and energy fluxes from three LSMs as a result of cross-walking table uncertainty. This study found that the spread in model outputs due to cross-walking uncertainty was higher than uncertainty due to the underlying LC maps (mapping algorithm) (Hartley et al., 2017). Despite these uncertainties, satellites provide the only plausible way to derive the global maps of vegetation distribution needed to drive LSMs and validate dynamic global vegetation models. Future efforts by the ESA CCI LC project and collaborators will

focus on reducing the uncertainty introduced when translating from LC to PFT, including using optimized and regionally-based cross-walking tables.

## 4.2 Differences in area changes

The ESA CCI LC magnitudes of gross changes for all PFT are lower than those of all three products considered. This is explained by the effect of spatial resolution combined with a change consolidation approach. Using Earth Observation time

series of 1 km spatial resolution to detect annually the land cover change for the ESA CCI maps does not allow capturing small scale LC changes, which is part of the reasons for smaller gross and net forest changes than those in Hansen et al. (2013). On the other hand, this is the only way to have a consistent method of LC change detection over the whole period. In spite of the consolidation strategy confirming the change over several years, ESA CCI LC trends of area change mitigate only partly the impact of the heterogeneous quality of the data acquired by the various sensors. For instance, larger change

variations for forest and cropland in the 1990s result from poorer radiometric and spectral quality of the AVHRR input data. This instrument, first designed for meteorological observation, is however the only one recording the land surface systematically before 1999.



The large magnitude of gross changes in forest and cropland in LUH2v2h (Hurtt et al., 2011) (Figure 2) is partly caused by the large shifting cultivation area in tropical Africa (Figure S3). The area of shifting agriculture is reduced from LUH1 to LUH2v2h (Figure S3) because of the separation of forest from natural vegetation in LUH2 (Hurtt et al., 2011). However, the gross forest changes in LUH2v2h (Hurtt et al., 2011) are still much higher than those in ESA CCI PFTs and Hansen et al.

(2013). Especially in the ESA 300m resolution data, the gross change area seems very small (Figure S3).

The discrepancies in temporal PFT net area changes between ESA CCI maps and FAO data (cropland and pasture area changes in LUH2v2h (Hurtt et al., 2011) and forest area changes in Houghton and Nassikas (2017), Figure 4-6) are mainly caused by the different approaches for estimating LC change used by different countries in FAO reports (FAO, 2015; FAOSTAT, 2015). Some countries like Canada distinguish land use and land cover when compiling forest statistics. For

example, a forest cleared for wood harvest is not taken as a forest loss because new secondary forest will be planted on this land, thus no change in land use. However, remote sensing can easily detect such land cover change and treat it as forest loss. Cropland and pasture in HYDE 3.2 (Klein Goldewijk et al., 2016) adopted the FAO categories for "Arable land and permanent crops" and "Permanent meadows and pastures" respectively as the main data source. In the ESA CCI LC maps, pastures are mapped as grassland and translated into 100 % "Natural Grass" PFT (Table S1). Finally, the trends of cropland

area change from FAO STAT data may contradict those from national statistics, e.g. comparing FAO STAT data (FAOSTAT, 2015) with USDA estimates (Nickerson et al., 2011) for United States or with NBSC estimates (NBSC, 2015) for China (Li et al., 2016).

## 5 Conclusions

In this study, we compare the absolute areas and areal changes between PFTs from annual ESA CCI LC products and other

datasets. In the intensive LULCC regions like South and Central America, both forest area and net forest change are consistent with those from other datasets. The detection of LC changes has significantly improved from the last version of five-year epoch ESA CCI maps (Li et al., 2016). The detailed annual cropland changes from 1992 to 2000 fill the gaps of HYDE 3.2 data for this period, in which only decadal changes are available (Klein Goldewijk et al., 2016).

Considering the discrepancies, advantages and defects among different datasets (Table 1), we propose different choices of

these datasets for the applications in LSMs depending on our research purposes. For example, if we would like all LSMs to share the same historical and future maps in a model intercomparison project (e.g. using LUH2v2h data in CMIP6), annual ESA CCI data products should be cautiously harmonized considering the large differences between ESA CCI and LUH2v2h (Hurtt et al., 2011). On the other hand, if we want to analyze recent carbon and water budgets with LSMs, ESA CCI maps are definitely an appropriate choice. The detailed LC classes in ESA CCI products provide a valuable reference map for

modellers to partition land covers into PFTs, e.g. separating the generic forest in LUH2v2h dataset (Hurtt et al., 2011) into different forest PFTs (Table S1). LSMs can also benefit from the 300m spatial resolution changes in ESA CCI products when accounting for gross land use changes to simulate the LULCC carbon fluxes. Therefore, the current annual ESA CCI





land cover maps with full land cover classes, 300 m spatial resolution and relatively long-time series are sufficient to be implemented in LSMs and help better characterize the recent global and regional carbon cycles.

## Acknowledgements

W.L. and P.C. were supported by the European Research Council through Synergy grant ERC-2013-SyG-610028 "IMBALANCE-P". N.M., S.B., C.L. and P.D were supported by the Climate Change Initiative program supported by the European Space Agency.

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




**Figure 1. Global and regional areas of forest, cropland and grassland PFTs in year 2000 in comparison with data from LUH2 v2h (Hurtt et al., 2011), Hansen et al. (2013) and Houghton and Nassikas (2017). Different colors indicate different PFTs.**

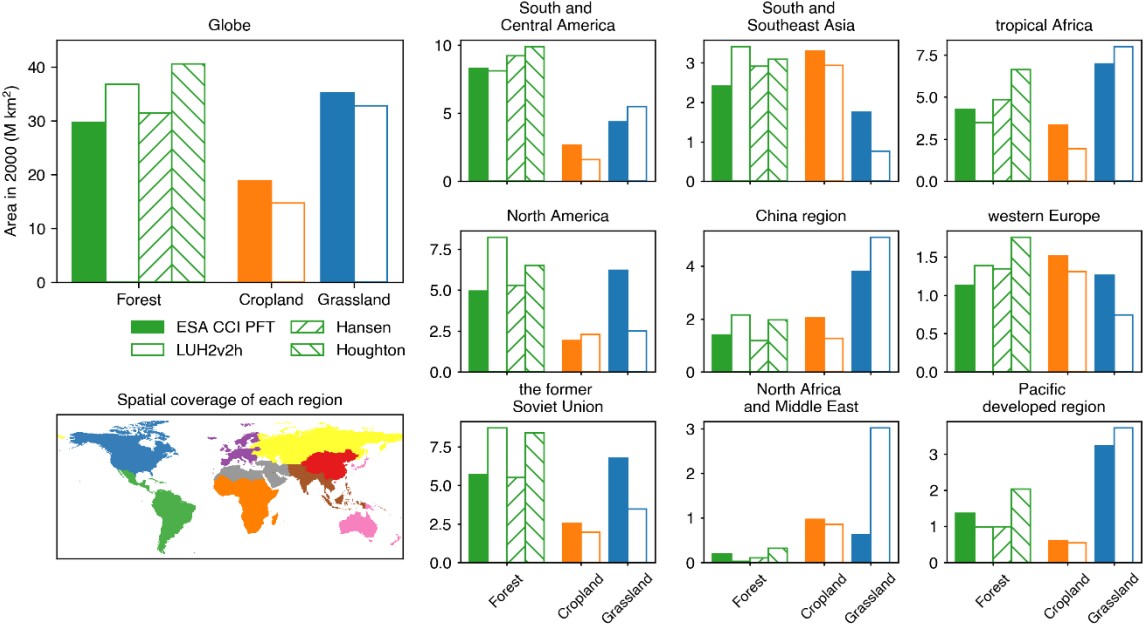



**Figure 2. Gross changes of PFTs from 1992 to 2015 after translating gross transitions between original ESA land cover classes. Gross changes from LUH2v2h (Hurtt et al., 2011) and Hansen et al. (2013) are also shown for comparison. The red line indicates the zero line.**

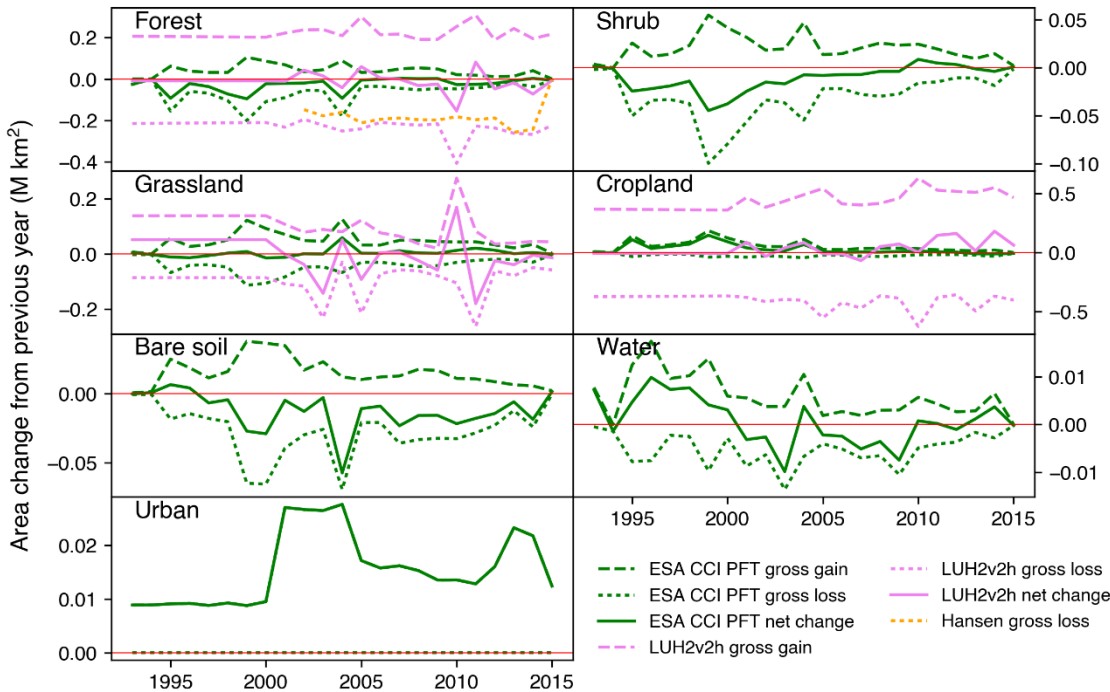





**Figure 3. Spatial distributions of net and cumulative gross changes of forest, cropland and grassland PFTs between 1992 and 2015 derived from the ESA CCI data. Color scale indicates the changed fraction in each half degree grid cell.**

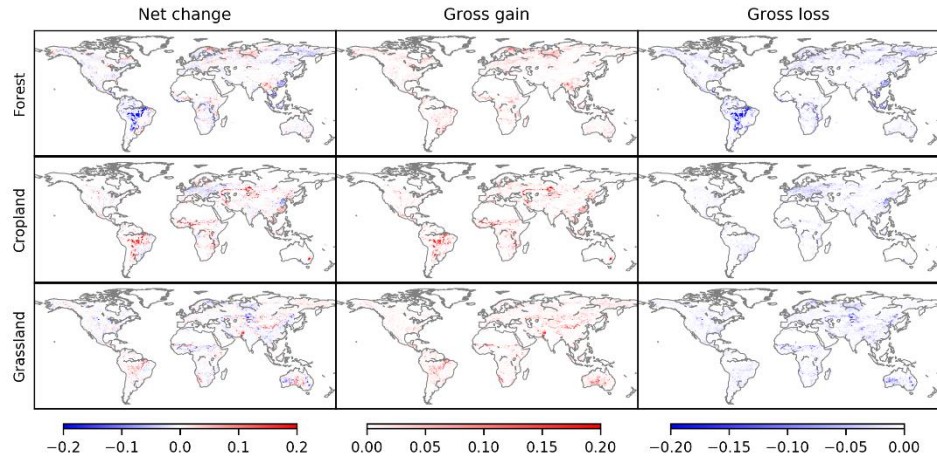





**Figure 4. Global and regional net area changes of forest, cropland and grassland PFTs derived from ESA CCI land cover maps since 1992. Data from LUH2v2h (Hurtt et al., 2011), Hansen et al. (2013) and Houghton and Nassikas (2017) are also shown for comparison. Note that net forest area change from Hansen et al. (2013) is corresponding to the period of 2000-2012, and thus the forest area change between1992 and 2000 from ESA CCI was added in Hansen et al. (2013) data in the plot.**

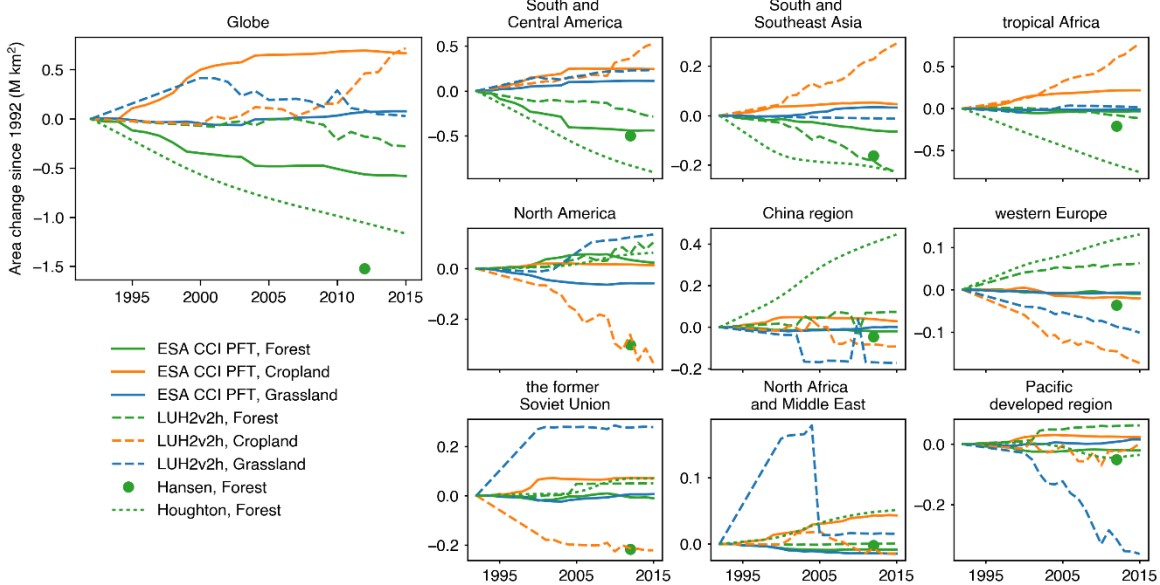



**Figure 5. Net area changes of forest, cropland and grassland PFTs derived from ESA CCI land cover maps since 1992 in countries with largest net forest area loss between 1992 and 2015. Data from LUH2v2h (Hurtt et al., 2011), Hansen et al. (2013) and Houghton and Nassikas (2017) are also shown for comparison.**

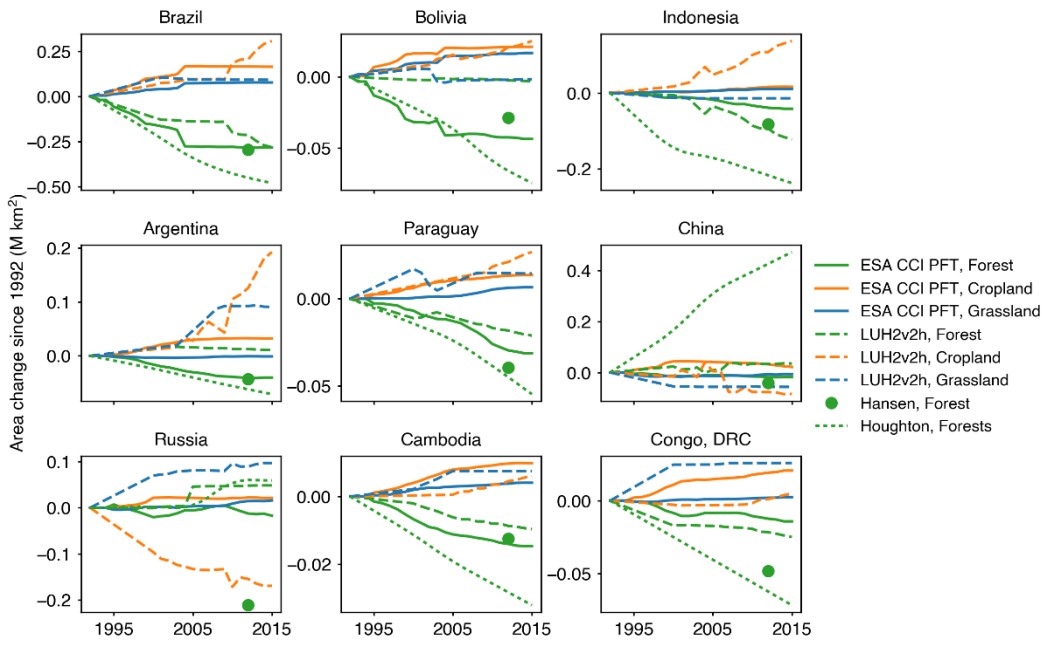

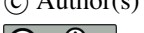



**Figure 6. Net area changes of forest, cropland and grassland PFTs derived from ESA CCI land cover maps since 1992 in countries with largest net forest area gain between 1992 and 2015. Data from LUH2v2h (Hurtt et al., 2011), Hansen et al. (2013) and Houghton and Nassikas (2017) are also shown for comparison.**

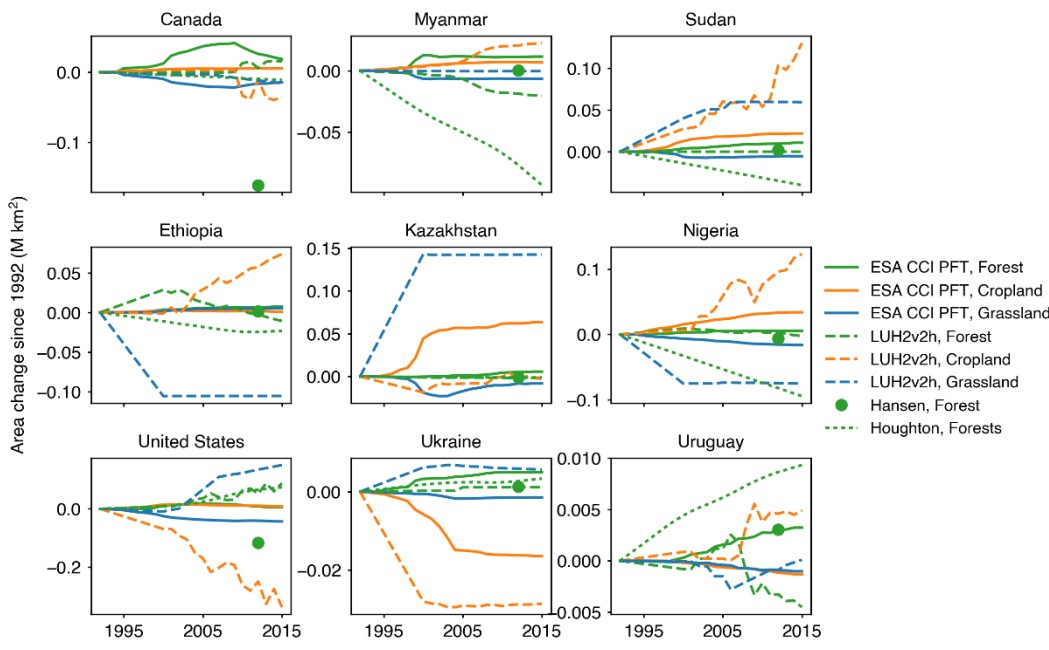



**Table 1. Description and comparison of different land-use / land-cover datasets used in this study.**

|  | PFTs from annual ESA CCI maps | LUH2v2h (Hurtt et al., 2011) | HYDE 3.2 (Klein Goldewijk et al., 2016) | Hansen et al. (2013) | Houghton and Nassikas (2017) |
|---|---|---|---|---|---|
| Time span | 1992-2015 | 850-2100 | 10,000 BC-2015 | 2000-2014 | 1850-2015 |
| Time step | annual | annual | 1000 yr for the BCE period, then 100 yr till 1700, 10 yr till 2000, and from 2000 - 2015 annual | gross loss, annual; gross gain for one period (2000-2012) | annual |
| Spatial resolution | 300 m | 0.25 ° | 5 arc-minute | 30 m | country |
| Land-use / land-cover type | forest, shrub, grassland, cropland, bare soil, water and urban | forest, cropland, pasture, rangeland, urban and non-forested | cropland, grazing lands and urban | forest | forest |
| Gross or net | gross and net | gross and net | net | gross and net | net at country level |
| Data source | satellite (MERIS, SPOT-VGT, AVHRR, and PROBA-V) | urban, cropland, pasture and rangeland from HYDE 3.2 (Klein Goldewijk et al., 2016); forest and transitions based on model | cropland and grazing land are based on the FAO categories for "Arable land and permanent crops" and "Permanent meadows and pastures" (FAOSTAT, 2015); Spatial distribution based on ESA CCI epoch LC map 2010 | satellite (Landsat) | FAO FRA (FAO, 2015), based on country reports |
| Advantage | full land cover types; relatively long time series; relatively high resolution; full gross transitions | full gross transitions; long time series | long-time series; inventory-based | high resolution | inventory-based |
| Defect | no specific pasture; uncertainty in cross-walking table | no separation of deciduous and evergreen forest; model-based forest areas; model-based temporal changes of historical cropland and grazing land (HYDE 3.2) | no forest; coarse time steps | short time period; no annual forest gain, but only for the whole period of 2000-2012; no other LC types | not grid-cell explicit; no other LC types; inconsistency of data sources and forest definitions between different countries |