# Peer review of "Gross and net land cover changes based on plant functional types derived from the annual ESA CCI land cover maps"

_Earth System Science Data, 2017_

## Referee Comment (RC1) · Anonymous Referee #1 · 22 Sep 2017

General comments

The discussion paper presents an interesting dataset (PFT annual maps at $0.5 \times 0.5$ deg resolution derived from Global Land Cover annual maps) which should be made publically available as it is the core dataset of this study.

The paper focuses mainly on the comparison of estimates of areas, gross and net changes of different plant functional types (maps of PFTs derived from the ESA CCI LC product) with 3 other sources : Hurtt et al. (2011), Hansen et al. (2013) and Houghton and Nassikas (2017). This comparison is useful for understanding the range of discrepancies between such datasets.

The datasets and the results of the comparison are clearly expressed and well presented. However the discussion should be complemented with further issues which can explain part of the discrepancies between the CCI LC derived PFT dataset and independent datasets (see section 'Comparison with other datasets' in Specific Comments for detailed information on such issues).

Specific comments

Title:

I suggest to revise the title in order to relate better to the content of paper, e.g. : "Gross and net land cover changes of the main plant functional types derived from the annual ESA CCI land cover maps (1992-2015)"

Access to the core dataset of this study:

In abstract: "The annual ESA CCI land cover products can be downloaded from http://maps.elie.ucl.ac.be/CCI/viewer/download.php" This paper is focused on the derivation of PFT change estimates from the ESA CCI LC product. ("our analyses are based on the PFT maps that have been translated from the ESA CCI LC maps, rather than the original LC classes"). Only one "example of LC map and PFT map in 2000 used in this study" is made available ("can be downloaded from doi: https://doi.org/10.5281/zenodo.834229")

I consider that it would be more appropriate and pertinent for this paper to provide access to the full derived dataset (PFT annual maps at 0.5 × 0.5deg resolution) as main product of this study - in complement to the ESA CCI LC product which is already available through ESA and UCL web sites.

Use of FAO data:

FAO data are referred a few times in the paper, e.g. in Introduction: "Global net LULCC carbon emissions (ELUC) are estimated to be 1.1 ±0.4 Pg C yr-1 during the past decade (2006-2015) by the bookkeeping model of Houghton and Nassikas (2017)

[Figure]

based on the national land cover data from Food and Agriculture Organization (FAO)"
It would be useful to indicate if it refers to FAO FRA 2015 data or the FAO STAT along
the paper.

When referring to FAO FRA 2015 estimates, Keennan et al (2015) reference
should be added because its reports the main findings of the FRA-2015. More-
over Keennan et al (2015) is used as key reference by the global environ-
mental scientific community (see manuscript in-press with BioScience available:
http://scientistswarning.forestry.oregonstate.edu/

Keenan R J et al (2015) Dynamics of global forest area: results from the FAO Global
Forest Resources Assessment 2015. Forest Ecology and Management 352:9–20

Description of the ESA CCI land cover products (section 2.1) and their accuracies:

The reference to Yang et al 2017 (ISPRS Journal of Photogrammetry and Remote
Sensing 125 (2017) 156–173) should be added e.g. in introduction and / or section 2.1.
Yang et al 2017 reports the "Accuracy assessment of seven global land cover datasets
over China" including two maps from previous version of CCI LC dataset (epochs 200
and 2010)

"The accuracy of ESA CCI LC products was evaluated at global scale. An object -
based validation database of 2600 Primary Sampling Units was built by a panel of
international experts to specifically assess the accuracy of both the LC classes and
change (ESA, 2017)." The estimates of accuracy of the ESA CCI LC products should
be provided here, based on published results in ESA (2017) report. It should also be
clearly mentioned that accuracy of changes was not evaluated / quantified.

Comparison with other datasets (section 2.4 and discussion section):

Keennan et al (2015) compares the findings of FRA 2015 with other remote sensing
studies. It provides some explanation and discussion on the issue raised by the author:
" land use data are not necessarily the same as land cover, and the exact definitions

and categorization of forest (cropland and grassland) are different for each dataset".

The discussion on differences in area and area changes (section 4.1. and 4.2) is interesting and covering a number of important issues, but it should be complemented by at least two further issues: A main difference between FAO FRA-2015 dataset and Hansen et al (2013) product is that FAO reports a land use definition when Hansen reports a Tree cover percentages. A major impact of such differences in definition is related to Oil Palm plantations. Oil Palm plantations are not reported as forests by FAO (considered as agricultural use) when they are mapped as dense Tree Cover by Hansen. This difference has major impacts on estimates of LC changes for countries like Indonesia. It would be useful to pay attention to this specific vegetation type and to mention the Land Cover class under which are mapped Oil Palm plantations in CCI LC product (regional class 'Tree or Scrub Cover' under 'Cropland' first level class) and to mention to which PFT it has been attributed (Forest or croplands).

It is also known in the remote sensing community that it is difficult to map and estimate forest areas in the dry tropics with medium resolution satellite (Landsat type), in particular when tree cover is below 40 %. Consequently it is at least as challenging or more difficult to map such forests from coarser resolution imagery or to estimate accurately area changes from medium or coarse resolution data. This is illustrated, reported or discussed in a number of papers including Hansen et al (2013) and Achard et al (2014) and more recently in Bastin et al (2017) and Gross et al (2017). This can explain partly why the estimate of forest area derived from the CCI LC product is lower than national estimates derived from the FAO FRA dataset.

Achard F et al (2014) Determination of tropical deforestation rates and related carbon losses from 1990 to 2010 Global Change Biology (2014) 20, 2540–2554

Gross D et al (2017) Uncertainties in tree cover maps of Sub-Saharan Africa and their implications for measuring progress towards CBD Aichi Targets. Remote Sens Ecol Conserv. doi:10.1002/rse2.52

[Figure]

The remote sensing community, in particular scientists dealing with monitoring of REDD+ activities, has produced technical guidelines or scientific papers which report that it is more efficient and accurate to produce area estimates by combining a sample of reference dataset (sample of reference plots) with a wall to wall map, than by using only a wall to wall map. This is particularly valid for estimating Land Cover changes which are usually considered as 'rare' events. See GOFC-GOLD 2015, GFOI 2016, Olofsson et, 2014; Sannier et al 2016

GOFC-GOLD, 2015, A Sourcebook of Methods and Procedures for Monitoring and Reporting Anthropogenic Greenhouse Gas Emissions and Removals Associated with Deforestation, Gains and Losses of Carbon Stocks in Forests Remaining Forests, and Forestation (GOFC-GOLD Land Cover Project Office, Wageningen University, The Netherlands).

GFOI 2016, Integration of remote-sensing and ground-based observations for estimation of emissions and removals of greenhouse gases in forests: Methods and Guidance from the Global Forest Observations Initiative, Edition 2.0, FAO, Rome

Olofsson P et al 2014, Good practices for estimating area and assessing accuracy of land change. Remote Sensing of Environment 148 (2014) 42–57

Sannier C et al 2016 Suitability of Global Forest Change data to report forest cover estimates at national level in Gabon. Remote Sensing of Environment, 173, 326-338

Figures to be improved

Figure 3 is much too small to see the changes

Figures 4 to 6 would also benefit to be displayed over a larger area.

---

## Referee Comment (RC2) · Anonymous Referee #2 · 6 Oct 2017

General comments: This paper investigated recent global land cover change (gross and net). The analysis based on recent annual remote sensing maps (ESA-CCI). The results of this study were compared with other data sets. The authors presented a nice data-driven analysis to assess gross land change dynamics, which serves as a valuable contribution for validating gross land change dynamics around the world, a necessity; given that, gross land changes have significant impact on our Earth System. Although the study itself was carried out very well, I have serious doubts in the quality of the input data used for the analysis. Before the paper can be published, a number of major issues should be tackled and clarified in the document first.

Major Comments:

First: Throughout the manuscript, the authors referred to data sets for comparison (Hansen 2013, Hurtt et al. 2011, Houghton & Nassakas (2017). I assume that people from various research disciplines will be interested in reading this paper. However, each one of them might consider something else as a data set. Data sets often refer to measurements (e.g. remote sensing), while you also list historic reconstructions, a model output, as data sets (e.g. Hurtt et al.). I would advise to make very clear what the differences are between measurements and reconstructed model outputs.

Second: The authors' main aim seemed to be the comparison of data from the observational period with reconstructed model outputs. I do not fully understand why the authors only used one data set (ESA-CCI) instead of using multiple data sets of the observational period, knowing that many other data sets would have been available for larger regions (U.S., Europe, China, Africa, India, Indonesia, Brazil, etc.) or even globally (Globeland30). This would have strengthened their observational evidence. The authors argument from page 5 (line 7), that their chosen data sets for comparison were the best data sets available does not really hold and seems artificial. These products are commonly known state-of-the art products for land cover and land use change, but not necessarily the best available to assess gross land changes. A critical reflection in the introduction and discussion section would be good to highlight alternatives (from both observations and model reconstructions).

Third: The authors described on page 3 (bottom) and page 4 (top) the accuracy assessment that was performed for ESA-CCI. I was wondering, what were the results? I could not find a single accuracy measure result. How does this product compare with others? Does it qualify to assess land cover change?

Fourth: Recently I reviewed a paper that compared the suitability of different observation based products for cropland monitoring. Compared to FAO cropland statistics and other observation based products (GLC2000, MODIS, GLC-Share, Geo-Wiki, GLC-

NMO2008, Globeland30) the ESA-CCI products (epoch maps and yearly maps) and the previous Globecover product seem to overestimate cropland by lot (20% and more compared to others). Unfortunately, the paper is still in review, otherwise I would have forwarded it. Other than discrepancies in definitions and spatial resolution, which were mentioned by the authors, I wonder how suitable the classification algorithm of ESA-CCI (and Globecover, since the same group carried it out) is for land cover detection. Reading these numbers, I have serious doubts. Your study seems to support these numbers: global forest area was underestimated by roughly 20-25% compared to other products (page 6 first paragraph), while cropland was overestimated by ca. 20% compared to Hurtt et al., which is based on FAO estimates in the end. Again, here I would like to see a critical discussion.

Fifth: This brings me from land cover detection to land cover change detection. The authors mentioned that all products used for comparison (Hurtt, Hansen, Houghton & Nassikas) yielded more gross land changes than ESA-CCI. To be honest, I am a bit puzzled. How can a model reconstruction (LUH2) that is largely based on net land changes (due to HYDE 3.2), which again is based on FAO net land changes, yield more gross change than RS-based products? LUH2 only accounts for gross land changes in shifting cultivation areas and it was proven that gross land changes also appear in other world regions (Fuchs et al. 2015 & 2016, Global Change Biology; Bayer et al. 2016, Earth System Dynamics). It seems that ESA-CCI is not optimal to detect land cover changes for various reasons. Differences in spatial resolution between products does not seem to play a role between Hurtt et al. and ESA-CCI. Again, a critical discussion is urgently needed.

Sixth: All changes were given as changes in km$^2$, spread throughout the document here and there. Personally, I find this hard to compare and put in relation. I would recommend a table with yearly change rates in percent (global and continental) for each of your products. This way a direct comparison per region and product is possible as helps the reader to find what he is looking for.

---

## Author Comment (AC1) · 9 Nov 2017

**Response to comments**

**Paper #:** *essd-2017-74*
**Title:** *Gross and net land cover changes based on plant functional types derived from the annual ESA CCI land cover maps*
**Journal:** *Earth System Science Data*

**Reviewer #1:**

**General Comments:**

**Comment #1**

The discussion paper presents an interesting dataset (PFT annual maps at $0.5 \times 0.5$ deg resolution derived from Global Land Cover annual maps) which should be made publically available as it is the core dataset of this study. The paper focuses mainly on the comparison of estimates of areas, gross and net changes of different plant functional types (maps of PFTs derived from the ESA CCI LC product) with 3 other sources : Hurtt et al. (2011), Hansen et al. (2013) and Houghton and Nassikas (2017). This comparison is useful for understanding the range of discrepancies between such datasets.

The datasets and the results of the comparison are clearly expressed and well presented. However the discussion should be complemented with further issues which can explain part of the discrepancies between the CCI LC derived PFT dataset and independent datasets (see section 'Comparison with other datasets' in Specific Comments for detailed information on such issues).

**Response #1**

We thank the reviewer for the comments and suggestions. Please see the detailed point-by-point responses below.

**Specific Comments:**

**Comment #2**

Title: I suggest to revise the title in order to relate better to the content of paper, e.g. : "Gross and net land cover changes of the main plant functional types derived from the annual ESA CCI land cover maps (1992-2015)"

**Response #2**

Revised as suggested.

**Comment #3**

Access to the core dataset of this study:

In abstract: "The annual ESA CCI land cover products can be downloaded from http://maps.elie.ucl.ac.be/CCI/viewer/download.php" This paper is focused on the derivation of PFT change estimates from the ESA CCI LC product. ("our analyses are based on the PFT maps that have been translated from the ESA CCI LC maps, rather than the original LC classes"). Only one "example of LC map and PFT map in 2000 used in this study" is made available ("can be downloaded from doi: https://doi.org/10.5281/zenodo.834229")

I consider that it would be more appropriate and pertinent for this paper to provide access to the full derived dataset (PFT annual maps at $0.5 \times 0.5$deg resolution) as main product of this study - in complement to the ESA CCI LC product which is already available through ESA and UCL web sites.

**Response #3**

We will upload the full derived dataset (annual PFT maps at $0.5° \times 0.5°$ resolution) to the data repository website and give a doi in the revised manuscript.

**Comment #4**

Use of FAO data:

FAO data are referred a few times in the paper, e.g. in Introduction: "Global net LULCC carbon emissions (ELUC) are estimated to be 1.1 ±0.4 Pg C yr-1 during the past decade (2006-2015) by the bookkeeping model of Houghton and Nassikas (2017) based on the national land cover data from Food and Agriculture Organization (FAO)" It would be useful to indicate if it refers to FAO FRA 2015 data or the FAO STAT along the paper.

When referring to FAO FRA 2015 estimates, Keennan et al (2015) reference should be added because its reports the main findings of the FRA-2015. Moreover Keennan et al (2015) is used as key reference by the global environmental scientific community (see manuscript in-press with BioScience available: http://scientistswarning.forestry.oregonstate.edu/)

Keenan R J et al (2015) Dynamics of global forest area: results from the FAO Global Forest Resources Assessment 2015. Forest Ecology and Management 352:9–20

**Response #4**

The national forest area data from Houghton and Nassikas (*2017*) are based on FAO FRA data. As suggested, we will note it in Introduction and Methods and add Keenan et al. (*2015*) as a reference accordingly. We will also add Keenan et al. (*2015*) in Discussion (see **Response #6**).

**P2L8**: "Global net LULCC carbon emissions ($E_{LUC}$) are estimated to be 1.1 ±0.4 Pg C yr$^{-1}$ during the past decade (2006-2015) by the bookkeeping model of Houghton and Nassikas (2017) based on the national land cover data from Food and Agriculture Organization Forest Resouces Assessment (FAO FRA) (FAO, 2015; Keenan et al., 2015)."

**P5L2**: "The national forest areas from Houghton and Nassikas (2017) are based on FAO Forest Resources Assessment (FRA) data (FAO, 2015; also see Keenan et al. (2015) for the main findings of FAO FRA 2015)."

**Comment #5**

Description of the ESA CCI land cover products (section 2.1) and their accuracies:

The reference to Yang et al 2017 (ISPRS Journal of Photogrammetry and Remote Sensing 125 (2017) 156–173) should be added e.g. in introduction and / or section 2.1. Yang et al 2017 reports the "Accuracy assessment of seven global land cover datasets over China" including two maps from previous version of CCI LC dataset (epochs 200 and 2010)

"The accuracy of ESA CCI LC products was evaluated at global scale. An object -based validation database of 2600 Primary Sampling Units was built by a panel of international experts to specifically assess the accuracy of both the LC classes and change (ESA, 2017)." The estimates of accuracy of the ESA CCI LC products should be provided here, based on published results in ESA (2017) report. It should also be clearly mentioned that accuracy of changes was not evaluated / quantified.

**Response #5**

We will cite the reference by Yang et al. (2017) in **Section 2.1** and add sentences about the accuracy of ESA CCI LC products on **P4L4**: "The accuracy of ESA CCI LC products was evaluated at global scale according to international standards, using an independent validation dataset to produce confusion matrix and derive overall accuracy figure. An object-based validation database of 2600 Primary Sampling Units was built by a panel of international experts to specifically assess the accuracy of both the LC classes and changes (ESA, 2017). Research is currently ongoing to find how addressing the new challenges underlying this database, i.e. following a per-object approach and interpreting not a unique land cover class but a distribution of land cover classes within a Primary Sampling Unit. The uniqueness of these two concepts in the framework of global land cover validation results that more time is needed to derive reliable figures about LC classes and LC changes accuracy. It will also prevent from any comparison with previous validation figures.

In this respect, for the sake of comparison, the accuracy of the ESA CCI LC product from 2010 was assessed using the GlobCover 2009 validation database (Bontemps et al. 2010). Using all the points

interpreted as "certain" by the experts, whether "homogeneous" (i.e. made of a single LC class) or "heterogeneous" (i.e. made of several or mosaic LC classes), the overall accuracy was found to be 71.5%. Accounting only the "homogeneous" and "certain" points, the overall accuracy raised to 75.4% (ESA, 2017). The highest user accuracy values were found for the classes of rainfed cropland, irrigated cropland, broadleaved evergreen forest, urban areas, bare areas, water bodies and permanent snow and ice. Conversely, mosaic classes of natural vegetation were associated with the lowest user accuracy values, as well as the three classes of lichens and mosses, sparse vegetation and flooded forest with fresh water.

The overall accuracy of the ESA CCI LC products was also assessed by independent studies over specific regions (e.g. Tsendbazar et al. (2015) over Africa and Yang et al. (2017) over China), which can give valuable insights for specific applications."

However, as we described on **P3L9**: "The objectives of this study are to document the major gross and net changes and transitions in PFT maps derived from annual ESA CCI LC products and to evaluate whether they can be used in LSMs.", we only focus on the translated PFT maps for the use of LSMs rather than the original ESA land cover classes. So, we didn't expand the accuracy assessment in this study.

**Reference:**

*Bontemps, S., Defourny, P., Van Bogaert, E., Kalogirou, V. and Arino, O., GlobCover 2009 - Products Description and Validation Report (2010). Available at: http://due.esrin.esa.int/page_globcover.php*

*Tsendbazar N.E., de Bruin S., Fritz S. and Herold M. 2017. Spatial Accuracy Assessment and Integration of Global Land Cover Datasets, Remote Sensing, 2015, 7(12), 15804-15821; doi:10.3390/rs71215804*

**Comment #6**

Comparison with other datasets (section 2.4 and discussion section):

Keenan et al (2015) compares the findings of FRA 2015 with other remote sensing studies. It provides some explanation and discussion on the issue raised by the author: "land use data are not necessarily the same as land cover, and the exact definitions and categorization of forest (cropland and grassland) are different for each dataset".

The discussion on differences in area and area changes (section 4.1. and 4.2) is interesting and covering a number of important issues, but it should be complemented by at least two further issues: A main difference between FAO FRA-2015 dataset and Hansen et al (2013) product is that FAO reports a land use definition when Hansen reports a Tree cover percentages. A major impact of such differences in definition is related to Oil Palm plantations. Oil Palm plantations are not reported as forests by FAO (considered as agricultural use) when they are mapped as dense Tree Cover by Hansen. This difference has major impacts on estimates of LC changes for countries like Indonesia. It would be useful to pay attention to this specific vegetation type and to mention the Land Cover class under which are mapped Oil Palm plantations in CCI LC product (regional class 'Tree or Scrub Cover' under 'Cropland' first level class) and to mention to which PFT it has been attributed (Forest or croplands).

**Response #6**

We will cite Keenan et al. (*2015*) when discussing the differences between remote sensing data and FAO FRA data on **P10L15** "Furthermore, in the definition of forest by FAO, natural disturbance suppressing forests do not change the land remaining a forest, but from satellite, they are not detected as forest cover. Keenan et al. (2015) also compared the forest area from FAO FRA 2015 with remote sensing data and attributed their differences to five factors, the major one of which is the different definitions of "forest"." and **P12L10** "For example, a forest cleared for wood harvest is not taken as a forest loss because new secondary forest will be planted on this land, thus no change in land use (Keenan et al., 2015). However, remote sensing can easily detect such land cover change and treat it as forest loss."

We agree that different definitions of oil palm plantation in different datasets is important for the forest/cropland area difference in S.E. Asia. As suggested, we will add discussion on **P10L24**: "The attribution of oil palm plantations is an important factor for the differences in area changes between

different datasets, especially in Indonesia. Oil palm is taken as cropland rather than forest in the FAO definitions (FAOSTAT, 2015) but detected as tree covers from the remote sensing (Tropek et al., 2014; Carlson et al., 2012, 2013; Koh et al., 2011; Hansen et al., 2013), including in the CCI LC products. This partly explains that the larger cropland increase in LUH2v2h (Hurtt et al., 2011) and larger forest decrease in Houghton and Nassikas (2017) than those in ESA CCI PFTs and Hansen et al. (2013) in Indonesia (Figure 4).".

**Reference:**

*Tropek, R., Sedláček, O., Beck, J., Keil, P., Musilová, Z., Símová, I. and Storch, D.: Comment on "High-resolution global maps of 21st-century forest cover change";., Science, 344(6187), 981, doi:10.1126/science.1248753, 2014.*

*Carlson, K. M., Curran, L. M., Ratnasari, D., Pittman, A. M., Soares-Filho, B. S., Asner, G. P., Trigg, S. N., Gaveau, D. A., Lawrence, D. and Rodrigues, H. O.: Committed carbon emissions, deforestation, and community land conversion from oil palm plantation expansion in West Kalimantan, Indonesia, Proc. Natl. Acad. Sci. USA, 109(19), 7559–7564, doi:10.1073/pnas.1200452109, 2012.*

*Carlson, K. M., Curran, L. M., Asner, G. P., Pittman, A. M., Trigg, S. N. and Adeney, J. M.: Carbon emissions from forest conversion by Kalimantan oil palm plantations, Nat. Clim. Chang., 3(3), 283–287, doi:Doi 10.1038/Nclimate1702, 2013.*

*Koh, L. P., Miettinen, J., Liew, S. C. and Ghazoul, J.: Remotely sensed evidence of tropical peatland conversion to oil palm, Proc. Natl. Acad. Sci., 108(12), 5127–5132, doi:10.1073/pnas.1018776108, 2011.*

**Comment #7**

It is also known in the remote sensing community that it is difficult to map and estimate forest areas in the dry tropics with medium resolution satellite (Landsat type), in particular when tree cover is below 40%. Consequently it is at least as challenging or more difficult to map such forests from coarser resolution imagery or to estimate accurately area changes from medium or coarse resolution data. This is illustrated, reported or discussed in a number of papers including Hansen et al (2013) and Achard et al (2014) and more recently in Bastin et al (2017) and Gross et al (2017). This can explain partly why the estimate of forest area derived from the CCI LC product is lower than national estimates derived from the FAO FRA dataset.

Achard F et al (2014) Determination of tropical deforestation rates and related carbon losses from 1990 to 2010 Global Change Biology (2014) 20, 2540–2554

Gross D et al (2017) Uncertainties in tree cover maps of Sub-Saharan Africa and their implications for measuring progress towards CBD Aichi Targets. Remote Sens Ecol Conserv. doi:10.1002/rse2.52

The remote sensing community, in particular scientists dealing with monitoring of REDD+ activities, has produced technical guidelines or scientific papers which report that it is more efficient and accurate to produce area estimates by combining a sample of reference dataset (sample of reference plots) with a wall to wall map, than by using only a wall to wall map. This is particularly valid for estimating Land Cover changes which are usually considered as 'rare' events. See GOFC-GOLD 2015, GFOI 2016, Olofsson et, 2014; Sannier et al 2016

GOFC-GOLD, 2015, A Sourcebook of Methods and Procedures for Monitoring and Reporting Anthropogenic Greenhouse Gas Emissions and Removals Associated with Deforestation, Gains and Losses of Carbon Stocks in Forests Remaining Forests, and Forestation (GOFC-GOLD Land Cover Project Office, Wageningen University, The Netherlands).

GFOI 2016, Integration of remote-sensing and ground-based observations for estimation of emissions and removals of greenhouse gases in forests: Methods and Guidance from the Global Forest Observations Initiative, Edition 2.0, FAO, Rome

Olofsson P et al 2014, Good practices for estimating area and assessing accuracy of land change. Remote Sensing of Environment 148 (2014) 42–57

Sannier C et al 2016 Suitability of Global Forest Change data to report forest cover estimates at national level in Gabon. Remote Sensing of Environment, 173, 326-338

**Response #7**

We thank the reviewer for this useful information. As suggested, we will add sentences to discuss the forest estimate in the arid region on **P10L11**: "In the drylands like tropical Africa, it is difficult to map and estimate forest area using medium (e.g. Landsat, Hansen et al., 2013) or coarse resolution satellite data (e.g. ESA CCI LC) (Bastin et al., 2017; Achard et al., 2014; Gross et al., 2017), in particular when tree cover is below 30% (Achard et al., 2014). Bastin et al. (2017) recently reported a forest estimate in drylands using very high spatial resolution satellite imagery, which is 40-47% more than previous forest assessments. The difficulty of detecting forest in these sparse tree cover regions could partly be responsible for the lower forest area from ESA CCI PFT maps than those from Hansen et al. (2013) and Houghton and Nassikas (2017) in tropical Africa (Figure 1)."

We will also add some discussion on the joint use of different datasets on **P12L18**: "In addition, instead of using a single dataset, combining a sample of several datasets is reported to be considerably more efficient and accurate to estimate land cover area and change (Olofsson et al., 2014; Sannier et al., 2016) and has been adopted as technical guidelines (GOFC-GOLD, 2015; GFOI, 2016) in the remote sensing community, especially for the forest monitoring in reduce emissions from deforestation and forest degradation in developing countries (REDD+) programme."

**Comment #8**

Figures to be improved

Figure 3 is much too small to see the changes

Figures 4 to 6 would also benefit to be displayed over a larger area.

**Response #8**

We will enlarge these figures in the revised manuscript and also upload the high-resolution figures separately.

---

## Author Comment (AC2) · 9 Nov 2017

**Response to comments**

**Paper #:** *essd-2017-74*
**Title:** *Gross and net land cover changes based on plant functional types derived from the annual ESA CCI land cover maps*
**Journal:** *Earth System Science Data*

**Reviewer #2:**

**General Comments:**

**Comment #1**

General comments: This paper investigated recent global land cover change (gross and net). The analysis based on recent annual remote sensing maps (ESA-CCI). The results of this study were compared with other data sets. The authors presented a nice data-driven analysis to assess gross land change dynamics, which serves as a valuable contribution for validating gross land change dynamics around the world, a necessity; given that, gross land changes have significant impact on our Earth System. Although the study itself was carried out very well, I have serious doubts in the quality of the input data used for the analysis. Before the paper can be published, a number of major issues should be tackled and clarified in the document first.

**Response #1**

We thank the reviewer for the comments and suggestions. Please see the detailed point-by-point responses below.

**Comment #2**

Major Comments:

First: Throughout the manuscript, the authors referred to data sets for comparison (Hansen 2013, Hurtt et al. 2011, Houghton & Nassakas (2017). I assume that people from various research disciplines will be interested in reading this paper. However, each one of them might consider something else as a data set. Data sets often refer to measurements (e.g. remote sensing), while you also list historic reconstructions, a model output, as data sets (e.g. Hurtt et al.). I would advise to make very clear what the differences are between measurements and reconstructed model outputs.

**Response #2**

We showed the differences of datasets used in this study in **Table 1**, including the time span, resolution, data source etc. We are aware that these datasets are from a variety of sources like remote sensing (e.g. ESA CCI, *Hansen et al., 2013*), historical reconstructions from models (e.g. forest area in *Hurtt et al., 2011*) or ground-based inventory (e.g. Houghton & Nassakas (*2017*) based on FAO FRA reports). The reasons why we chose these datasets for comparisons is that they are commonly used by the land surface modelling and LULCC carbon emission communities. Again, we would like to emphasize that the objective of these study as described on **P3L9**: "The objectives of this study are to document the major gross and net changes and transitions in PFT maps derived from annual ESA CCI LC products and to evaluate whether they can be used in LSMs."

We listed LUH2v2h (*Hurtt et al., 2011*) as a "dataset" because it is not purely model outputs and this "dataset" is well recognized and extensively used by the land surface modeling community, e.g. for the updates of global carbon budget by Le Quéré et al. (*2016*). As we described on **P4L30**: "The cropland and pasture areas in LUH2v2h dataset are from HYDE3.2 (Klein Goldewijk et al., 2016), in which ESA CCI epoch LC map in 2010 (representing 2008-2012) was used as a spatial reference map for the area allocation and the national cropland and grazing land were adjusted to match the FAO STAT data (FAOSTAT, 2015) as close as possible.". So, the cropland and pasture areas in LUHv2h are essentially based on satellite data and inventories. In addition, the wood harvest data from LUH2v2h (*Hurtt et al., 2011*) are also based on FAO inventory data.

We will further clarify these points (e.g. adding "The term of "datasets" in this study can also involve some model output (e.g. forest area from LUH2v2h)." in the caption of **Table 1**) and make clear differences between these datasets in the revised manuscript.

**Reference:**

*Le Quere, C., Andrew, R. M., Canadell, J. G., Sitch, S., Ivar Korsbakken, J., Peters, G. P., Manning, A. C., Boden, T. A., Tans, P. P., Houghton, R. A., Keeling, R. F., Alin, S., Andrews, O. D., Anthoni, P., Barbero, L., Bopp, L., Chevallier, F., Chini, L. P., Ciais, P., Currie, K., Delire, C., Doney, S. C., Friedlingstein, P., Gkritzalis, T., Harris, I., Hauck, J., Haverd, V., Hoppema, M., Klein Goldewijk, K., Jain, A. K., Kato, E., Kortzinger, A., Landschutzer, P., Lefevre, N., Lenton, A., Lienert, S., Lombardozzi, D., Melton, J. R., Metzl, N., Millero, F., Monteiro, P. M. S., Munro, D. R., Nabel, J. E. M. S., Nakaoka, S. I., O'Brien, K., Olsen, A., Omar, A. M., Ono, T., Pierrot, D., Poulter, B., Rodenbeck, C., Salisbury, J., Schuster, U., Schwinger, J., Seferian, R., Skjelvan, I., Stocker, B. D., Sutton, A. J., Takahashi, T., Tian, H., Tilbrook, B., Van Der Laan-Luijkx, I. T., Van Der Werf, G. R., Viovy, N., Walker, A. P., Wiltshire, A. J. and Zaehle, S.: Global Carbon Budget 2016, Earth Syst. Sci. Data, 8(2), 605–649, doi:10.5194/essd-8-605-2016, 2016.*

**Comment #3**

Second: The authors' main aim seemed to be the comparison of data from the observational period with reconstructed model outputs. I do not fully understand why the authors only used one data set (ESA-CCI) instead of using multiple data sets of the observational period, knowing that many other data sets would have been available for larger regions (U.S., Europe, China, Africa, India, Indonesia, Brazil, etc.) or even globally (Globeland30). This would have strengthened their observational evidence. The authors argument from page 5 (line 7), that their chosen data sets for comparison were the best data sets available does not really hold and seems artificial. These products are commonly known state-of-the art products for land cover and land use change, but not necessarily the best available to assess gross land changes. A critical reflection in the introduction and discussion section would be good to highlight alternatives (from both observations and model reconstructions).

**Response #3**

As we described on **P3L9**: "The objectives of this study are to document the major gross and net changes and transitions in PFT maps derived from annual ESA CCI LC products and to evaluate whether they can be used in LSMs." So, we didn't aim to compare "data from the observational period with reconstructed model outputs". The objectives are 1) provide the PFT maps from ESA CCI LC products to the land surface modelling community and 2) compare them with other commonly used land cover and land use maps in this community. For the land carbon modelling like the TRENDY project (http://dgvm.ceh.ac.uk/node/21/) for annual global carbon budget updates (*Le Quéré et al., 2016*), the land cover and land use change maps must have a global coverage and relatively long, consecutive and consistent time series. That's why we chose these datasets for comparisons in our study and excluding some regional maps or epoch maps (e.g. only 2000 and 2010 maps from Globeland30 are available). In fact, there have already been many studies on the detailed comparisons of different datasets in a region (e.g. *Fuchs et al. 2015* for Europe, *Yang et al. 2017* for China and *Achard et al. 2014* for tropics). In addition, we didn't plan to fully evaluate the accuracy of the original land cover detection from ESA satellites but focus on the translated PFTs that can be readily used by land surface models.

We will revise the sentence on **P5L7**: "Nevertheless, these represent the best datasets available for the use in LSMs for comparison…".

As suggested, we will highlight these alternative datasets in Discussion on **P12L19**: "There are also many other land cover and land use datasets that can be used for comparisons to assess the accuracy of land cover or land cover change in ESA CCI LC products. However, they are either regional maps (e.g. the maps for Europe from Fuchs et al., 2015) or global epoch maps (e.g. the Globeland30 maps for 2000 and 2010, Chen et al., 2014) and not suitable for the application in LSMs. Thus, we didn't include them in this study. In fact, there have already been studies on the detailed comparisons of different datasets in a region (e.g. Fuchs et al. 2015 for Europe, Yang et al. 2017 for China and Achard et al. 2014 for tropics). In addition to the accuracy assessments conducted in ESA CCI project

(ESA, 2017), a systematic comparison with all other land cover datasets in future will help to validate the land cover classification and land cover change detection in the ESA CCI LC products."

**Reference:**

Achard, F., Beuchle, R., Mayaux, P., Stibig, H.-J., Bodart, C., Brink, A., Carboni, S., Desclée, B., Donnay, F., Eva, H. D., Lupi, A., Raši, R., Seliger, R. and Simonetti, D.: *Determination of tropical deforestation rates and related carbon losses from 1990 to 2010, Global Chang. Biol., 20(8), 2540–2554, doi:10.1111/gcb.12605, 2014.*

Chen, J., Ban, Y. and Li, S.: *China: Open access to Earth land-cover map, Nature, 514(7523), 434–434, doi:10.1038/514434c, 2014.*

Fuchs, R., Herold, M., Verburg, P. H., Clevers, J. G. P. W. and Eberle, J.: *Gross changes in reconstructions of historic land cover/use for Europe between 1900 and 2010, Global Chang. Biol., 21(1), 299–313, doi:10.1111/gcb.12714, 2015.*

Yang, Y., Xiao, P., Feng, X. and Li, H.: *Accuracy assessment of seven global land cover datasets over China, ISPRS J. Photogramm. Remote Sens., 125, 156–173, doi:10.1016/j.isprsjprs.2017.01.016, 2017.*

**Comment #4**

Third: The authors described on page 3 (bottom) and page 4 (top) the accuracy assessment that was performed for ESA-CCI. I was wondering, what were the results? I could not find a single accuracy measure result. How does this product compare with others? Does it qualify to assess land cover change?

**Response #4**

We will add sentences on **P4L4**: "The accuracy of ESA CCI LC products was evaluated at global scale according to international standards, using an independent validation dataset to produce confusion matrix and derive overall accuracy figure. An object-based validation database of 2600 Primary Sampling Units was built by a panel of international experts to specifically assess the accuracy of both the LC classes and changes (ESA, 2017). Research is currently ongoing to find how addressing the new challenges underlying this database, i.e. following a per-object approach and interpreting not a unique land cover class but a distribution of land cover classes within a Primary Sampling Unit. The uniqueness of these two concepts in the framework of global land cover validation results that more time is needed to derive reliable figures about LC classes and LC changes accuracy. It will also prevent from any comparison with previous validation figures.

In this respect, for the sake of comparison, the accuracy of the ESA CCI LC product from 2010 was assessed using the GlobCover 2009 validation database (Bontemps et al. 2010). Using all the points interpreted as "certain" by the experts, whether "homogeneous" (i.e. made of a single LC class) or "heterogeneous" (i.e. made of several or mosaic LC classes), the overall accuracy was found to be 71.5%. Accounting only the "homogeneous" and "certain" points, the overall accuracy raised to 75.4% (ESA, 2017). The highest user accuracy values were found for the classes of rainfed cropland, irrigated cropland, broadleaved evergreen forest, urban areas, bare areas, water bodies and permanent snow and ice. Conversely, mosaic classes of natural vegetation were associated with the lowest user accuracy values, as well as the three classes of lichens and mosses, sparse vegetation and flooded forest with fresh water.

The overall accuracy of the ESA CCI LC products was also assessed by independent studies over specific regions (e.g. Tsendbazar et al. (2015) over Africa and Yang et al. (2017) over China), which can give valuable insights for specific applications. "

However, as we described on **P3L9**: "The objectives of this study are to document the major gross and net changes and transitions in PFT maps derived from annual ESA CCI LC products and to evaluate whether they can be used in LSMs.", we only focus on the translated PFT maps for the use of LSMs rather than the original ESA land cover classes. So, we didn't expand the accuracy assessment in this study.

**Reference:**

Bontemps, S., Defourny, P., Van Bogaert, E., Kalogirou, V. and Arino, O., *GlobCover 2009 - Products Description and Validation Report (2010). Available at: http://due.esrin.esa.int/page_globcover.php*

*Tsendbazar N.E., de Bruin S., Fritz S. and Herold M. 2017. Spatial Accuracy Assessment and Integration of Global Land Cover Datasets, Remote Sensing, 2015, 7(12), 15804-15821; doi:10.3390/rs71215804*

**Comment #5**

Fourth: Recently I reviewed a paper that compared the suitability of different observation based products for cropland monitoring. Compared to FAO cropland statistics and other observation based products (GLC2000, MODIS, GLC-Share, Geo-Wiki, GLC-NMO2008, Globeland30) the ESA-CCI products (epoch maps and yearly maps) and the previous Globecover product seem to overestimate cropland by lot (20% and more compared to others). Unfortunately, the paper is still in review, otherwise I would have forwarded it. Other than discrepancies in definitions and spatial resolution, which were mentioned by the authors, I wonder how suitable the classification algorithm of ESACCI (and Globecover, since the same group carried it out) is for land cover detection. Reading these numbers, I have serious doubts. Your study seems to support these numbers: global forest area was underestimated by roughly 20-25% compared to other products (page 6 first paragraph), while cropland was overestimated by ca. 20% compared to Hurtt et al., which is based on FAO estimates in the end. Again, here I would like to see a critical discussion.

**Response #5**

We thank the reviewer for this information. If we were right, the paper that the reviewer mentioned might be the one from JRC team entitled "*Comparison of global land cover datasets for cropland monitoring*" (*Pérez-Hoyos et al., 2017*) which is published now. This is correct to say that this paper does not conclude that the ESA CCI LC products are not the best suitable ones for cropland monitoring. We fully agree with the statement that the epoch-based ESA CCI LC products (NOT the one used in our study) were of lower quality for the cropland. This was recognized by the group generating the products, and specific effort was done to improve the mapping of this cropland class in the next version (i.e. 24 annual maps, the ones we used in our study). These efforts were recognized explicitly in the paper by Pérez-Hoyos et al. (*2017*) in varying sections:

- Section 4.1 about agreement between datasets: "*[2015 annual product] entails an important reduction of cropland in the Congo Basin zone*", meaning that the new 2015 annual product corrects errors in the 2010 epoch.

- Section 4.2 about agreement with FAO statistics: "*in America the best fit is found for LC-CCI2015 with 29.3 Mha compared to the 27.4 Mha of the FAO statistics*". In Asian and African countries considered in the paper, the ESA CCI-LC 2015 annual product is not the closest to the FAO statistics but the paper does not allow concluding that it is significantly lower than other global remote sensing product such as the MODIS one.

In the same section, Pérez-Hoyos et al. (*2017*) propose a figure showing that the "best fit" is really country-dependent and in this respect, the performance of the LC-CCI2015 is not systematically bad. This figure (*Pérez-Hoyos et al., 2017*) is included here below for the discussion.

[Figure]

**Figure 2.** Land cover dataset closest to FAOSTAT cropland area at the country level using the 100% weight.

- In the Section 4.3, Pérez-Hoyos et al. (*2017*) perform their own accuracy assessment of the different global land cover datasets. This is true that this section is not in favor of the ESA CCI-LC 2015 product, although the results are still contrasted depending on the region (ESA CCI-LC being at the 2nd position for American countries).

Pérez-Hoyos et al. (*2017*) conclude that in Africa, the products most suitable for agriculture monitoring are the GlobeLand30, which is at 30m spatial resolution, and the FAO-GLCshare which is an hybrid product integrating the best existing land cover datasets by ranking them based on specific priority criteria. About this latter product, they also explain that its suitability is mainly in "*countries where high resolution datasets are used*". This comes done to understanding that the spatial resolution is a key driver of products suitability for agriculture monitoring. This is certainly valid for this specific application, but it does not allow generalizing the conclusion to all domains.

In the American and Asian countries, Pérez-Hoyos et al. (*2017*) discuss that "*the advantage of GlobeLand30 and FAO-GLCshare is less evident*" and that they can only make country-specific recommendations, in which the ESA CCI-LC 2015 product does not perform less than other ones.

They also mention that "*Geowiki hybrid dataset is generally suitable*" but "*contains some spatial incoherencies (abrupt transitions) in some countries*", meaning that characteristics other than accuracy should also be considered.

Finally, we would like to draw attention to the fact that Pérez-Hoyos et al. (*2017*) did not at all consider the temporal dimension of the ESA CCI LC products in their assessment. Yet, they recognize that "*since the general interest in crop monitoring relies on the current crop distribution, a higher weight should be assigned to more recent reference data. In this sense, LC-CCI 2015 would have fewer divergences due to differences in time acquisition. Moreover, this product provides a time-series of yearly land cover maps that can been suitable for deriving change flows, but this has not yet been properly tested to our knowledge. On the contrary, this would penalize older layers, in particular GLC2000, but this is fair taking into account the purpose of this analysis*".

Accordingly, we don't understand the conclusions of Pérez-Hoyos et al. (*2017*) paper as being a reject of the ESA CCI LC products. Pérez-Hoyos et al. (*2017*) point out weaknesses that we should surely take into consideration but they also balance their conclusions depending on the country. We also have to keep in mind the specific framework of their paper: suitability for agriculture monitoring for early warning and the focus on a limited number of countries selected to be "*with high risk of food insecurity*". This second aspect is fully justified for a paper addressing the challenge of cropland

monitoring, but not the main target of a global land cover dataset, which in this case is used for deriving PFT maps for using in global land surface modeling. North America, Europe, Russia, Central Asia and Amazon Basin are for instance not considered, while they can be of significant importance for other applications. We should therefore keep in mind that this paper does not allow concluding about the reliability of the product at global scale.

Overall, we disagree with the reviewer that the differences in estimates may call into question the suitability of the ESA CCI algorithm. Rather, it is clear from the above discussion and other papers that compare datasets that issues of spatial resolution, LC class definitions, and the purpose of each dataset/study must be considered when comparing estimates. It is not clear from any of these studies which dataset should be seen as "truth". These issues need to be investigated further before the community can agree upon cropland extent. We also wish to point out that neither discerning the differences between datasets nor determining the validity of the extent of any particular biome, was the aim of this paper. We aim instead to assess the changes in vegetation distribution at the level of PFTs with the aim of using these maps to derive current and historical global PFT changes to drive land surface models, rather than specific purposes such as cropland monitoring. The differences between datasets will be discussed more widely in an upcoming ESA CCI Project paper. But we agree with the reviewer that we can discuss the cropland issues more fully in the discussion. We will therefore insert the following text after **P10L24:** "

Similarly, the underestimate in cropland area is likely due to differences in definitions of what constitutes a cropland based on remote sensing datasets used to derive the ESA maps versus land use statistics and country-dependent reporting used to derive FAO statistics that are used to define croplands in HYDE3.2 (Klein Goldewijk et al., 2016), in addition to differences in spatial resolution. First, the attribution of oil palm plantations is an important factor for the differences in area changes between different datasets, especially in Indonesia. Oil palm is taken as cropland rather than forest in the FAO definitions (FAOSTAT, 2015) but detected as tree covers from the remote sensing (Tropek et al., 2014; Carlson et al., 2012, 2013; Koh et al., 2011; Hansen et al., 2013), including in the CCI LC products. This partly explains that the larger cropland increase in LUH2v2h (Hurtt et al., 2011) and larger forest decrease in Houghton and Nassikas (2017) than those in ESA CCI PFTs and Hansen et al. (2013) in Indonesia (Figure 4). Second, the classification of cropland in ESA CCI is also based on remote sensing temporal analysis. In the ESA CCI algorithm, for example, spectral features at key moments during the year were used to optimize the discriminations between all major crop classes: differentiating between cropland and natural vegetation (typically harvesting dates). Cropland in LUH2v2h that is essentially from FAO statistics (Klein Goldewijk et al., 2016), on the other hand, depends on country reporting and therefore comprises different definitions and data sources from different countries.

Pérez-Hoyos et al. (2017) provide an extensive comparison of multiple cropland datasets, including ESA CCI epoch and annual maps, for the purposes of cropland monitoring, and they found that the ESA CCI 2015 annual map is more suitable for cropland monitoring than the epoch map because of the reduction in cropland area over the Congo basin. They also showed the spatial resolution is a key driver of products suitability for agriculture monitoring (Pérez-Hoyos et al., 2017). However, the specific framework of their study is the suitability for agriculture monitoring for early warning and the focus on a limited number of countries selected to be "*with high risk of food insecurity*" (Pérez-Hoyos et al., 2017). The issues of cropland area from the ESA CCI LC maps discussed in their study is fully justified for a study addressing the challenge of cropland monitoring, but it does not allow generalizing the conclusion to all domains (e.g. to derive PFT maps for using in global land surface modeling in this study). As Pérez-Hoyos et al. (*2017*) and our study shows, the agreement (or lack thereof) is country-dependent, further implying that more consistent definitions of LC classes are required and/or regional LC satellite mapping algorithms (or cross-walking table, see below) are needed. Cropland mapping issues, including those discussed in Pérez-Hoyos et al. (2017) are being addressed in upcoming versions of the ESA CCI maps. Additionally, Waldner et al. (2016) have produced a product that aims to combine the "fittest" LC maps at country level into a unified 250m cropland product, but again this is dependent upon a specific definition (the JECAM (Joint

Experiment of Crop Assessment and Monitoring) cropland definition for the purposes of cropland monitoring).

".

The differences in cropland areas could also be caused by the cross-walking table, and we discussed it on **P11L9**: "Likewise, an explicit regional classification is required for cropland. For example, class "10" (cropland, rainfed) is separated well in North America, i.e., mainly partitioning into class "11" (herbaceous cover), and thus the cropland area in this region is highly consistent with LUH2v2h data (Hurtt et al., 2011) (Figure 1). In tropical Africa where class "10" is not separated into a more detailed classification, the difference in cropland areas between these two datasets are large (Figure 1). This is because if most of the cropland in this region belongs to class "12", using the corresponding value for class "10" in the cross-walking table (90% for class "10" vs. 30% for class "12", Table S1) overestimates cropland areas."

In addition to the absolute cropland area, we will add some discussion on the difference in temporal cropland changes on **P12L14**: "The different trajectories of temporal cropland changes between ESA CCI and LUH2v2h (the former shows increasing from 1992 to 2004 while the latter increases after 2007, Figure 4) are probably caused by the time lag between the real changes and country reporting to FAO."

**Reference:**

P érez-Hoyos, A., Rembold, F., Kerdiles, H. and Gallego, J.: Comparison of Global Land Cover Datasets for Cropland Monitoring, Remote Sens., 9(11), 1118, doi:10.3390/rs9111118, 2017.

Waldner, F., Fritz, S., Di Gregorio, A., Plotnikov, D., Bartalev, S., Kussul, N., Gong, P., Thenkabail, P., Hazeu, G., Klein, I., Löw, F., Miettinen, J., Dadhwal, V., Lamarche, C., Bontemps, S. and Defourny, P.: A Unified Cropland Layer at 250 m for Global Agriculture Monitoring, Data, 1(1), 3, doi:10.3390/data1010003, 2016.

**Comment #6**

Fifth: This brings me from land cover detection to land cover change detection. The authors mentioned that all products used for comparison (Hurtt, Hansen, Houghton & Nassikas) yielded more gross land changes than ESA-CCI. To be honest, I am a bit puzzled. How can a model reconstruction (LUH2) that is largely based on net land changes (due to HYDE 3.2), which again is based on FAO net land changes, yield more gross change than RS-based products? LUH2 only accounts for gross land changes in shifting cultivation areas and it was proven that gross land changes also appear in other world regions (Fuchs et al. 2015 & 2016, Global Change Biology; Bayer et al. 2016, Earth System Dynamics). It seems that ESA-CCI is not optimal to detect land cover changes for various reasons. Differences in spatial resolution between products does not seem to play a role between Hurtt et al. and ESA-CCI. Again, a critical discussion is urgently needed.

**Response #6**

We will add regional gross change figures (reproduced below) to further compare the difference between ESA CCI PFTs and LUH2v2h (Hurtt et al., 2011). We will also add some discussion regarding the larger gross land use changes in LUH2v2h than in ESA CCI PFT maps on **P12L1**: "The large magnitude of gross changes in forest and cropland in LUH2v2h (Hurtt et al., 2011) (Figure 2) mostly distributes in the tropical regions (Figure S3 and S4) where gross changes reflect shifting agriculture (Heinimann et al., 2017). The gross gain and loss of forest (or cropland) in tropics from LUH2v2h maintains a similar constant rate with other small variations (Figure S3 and S4). This is because that the gross changes in LUH2v2h are mainly generated from the shifting cultivation in tropics by assuming a turnover rate of 6.7% yr$^{-1}$ (i.e. a residence time of 15 yr) of all agricultural lands (Hurtt et al., 2011) and based on a spatial distribution map from Bulter (1980). The Bulter (1980) map is a hand-drawn map indicating presence or absence (no precise area or fraction) of both shifting cultivation and other non-shifting farming systems based on some regional studies and "general knowledge" (Heinimann et al., 2017). The estimate of shifting cultivation extent from LUHv2h (Hurtt et al., 2011) is thus highly uncertain because of the simple assumptions and the old reference map (representing 1960s-1970s) but strongly affects the gross land use change areas. Heinimann et al.

(2017) recently estimated the global extent of shifting cultivation visually using Landsat 30 m forest data and very high resolution satellite imagery from Bing and Google. They found that shifting cultivation area decreases over the last 40 to 50 years, in particular in Southeast Asia (Heinimann et al., 2017). This however is not reflected in LUH2v2h dataset (Figure S3 and S4). From LUH1 to LUH2v2h, The area of shifting agriculture is reduced (see an example in tropical Africa, Figure S5) because of the separation of forest from natural vegetation in LUH2 (Hurtt et al., 2011). However, the gross forest changes in LUH2v2h (Hurtt et al., 2011) are still much higher than those in ESA CCI PFTs and Hansen et al. (2013). Especially in the ESA 300m resolution data, the gross change area seems very small (Figure S3 to S5). Therefore, the shifting cultivation area in LUH2v2h may be overestimated due to 1) the binary (presence / absence) indication rather than a precise extent of shifting cultivation in Butler (1980) map and 2) no temporal change (missing the decreasing trend) of the reference map. Still, it should be noted that the coarse spatial resolution of ESA CCI products cannot detect small-scale LC changes, resulting in an underestimation of gross changes. The shifting cultivation today remains extensive and is very important for the land carbon modeling, but there are only very limited studies on the regional or national extent estimates (Heinimann et al., 2017). More research and developments in the mapping and change detection of shifting cultivation are urgently desired."

**Reference:**

*Butler, J. H.: Economic Geography: Spatial and Environmental Aspects of Economic Activity, John Wiley & Sons., 1980.*

*Heinimann, A., Mertz, O., Frolking, S., Egelund Christensen, A., Hurni, K., Sedano, F., Parsons Chini, L., Sahajpal, R., Hansen, M. and Hurtt, G.: A global view of shifting cultivation: Recent, current, and future extent, edited by B. Poulter, PLoS One, 12(9), e0184479, doi:10.1371/journal.pone.0184479, 2017.*

**Figure S3** Global and regional gross changes in forest from different datasets.

[Figure]

**Figure S4** Global and regional gross changes in cropland from different datasets.

[Figure]

**Comment #7**

Sixth: All changes were given as changes in km$^2$, spread throughout the document here and there. Personally, I find this hard to compare and put in relation. I would recommend a table with yearly change rates in percent (global and continental) for each of your products. This way a direct comparison per region and product is possible as helps the reader to find what he is looking for.

**Response #7**

As suggested, we will add a table to give annual change rates in percent for each products (reproduced on next page).

**Table R1** The mean annual gross and net change rates (% of area in the reference year) in different PFTs from different datasets. Positive values of net changes indicate area increase. The gross change rates is calculated from the sum of absolute loss and gain, and thus always positive. The mean annual rates are calculated for the period of 1992-2015 using year 1992 as a reference year for all datasets except for Hansen et al. (2012) with a period of 2000-2012 and a reference year of 2000.

| Dataset | China region | North America | South and Central America | western Europe | tropical Africa | the former Soviet Union | South and Southeast Asia | Pacific developed region | North Africa and Middle East | Total |
|---|---|---|---|---|---|---|---|---|---|---|
| ESA CCI PFT, Forest, Net | -0.1 | 0.0004 | -0.2 | -0.04 | -0.04 | -0.01 | -0.1 | -0.1 | -0.2 | -0.1 |
| ESA CCI PFT, Forest, Gross | 0.4 | 0.2 | 0.4 | 0.4 | 0.3 | 0.3 | 0.5 | 0.5 | 0.7 | 0.3 |
| ESA CCI PFT, Cropland, Net | 0.1 | 0.01 | 0.1 | -0.1 | 0.2 | 0.1 | 0.1 | 0.1 | 0.9 | 0.1 |
| ESA CCI PFT, Cropland, Gross | 0.6 | 0.1 | 0.2 | 0.4 | 0.4 | 0.2 | 0.4 | 0.2 | 1.8 | 0.3 |
| ESA CCI PFT, Grassland, Net | 0.01 | -0.04 | 0.1 | -0.02 | -0.01 | 0.01 | 0.1 | 0.1 | -0.4 | 0.01 |
| ESA CCI PFT, Grassland, Gross | 0.8 | 0.2 | 0.2 | 0.3 | 0.4 | 0.3 | 0.3 | 0.8 | 1.2 | 0.3 |
| LUH2V2H, Forest, Net | 0.9 | 0.2 | 1.4 | 0.4 | 2.5 | 0.1 | 5.6 | 0.6 | 0.8 | 1.2 |
| LUH2V2H, Forest, Gross | 0.9 | 0.2 | 1.4 | 0.4 | 2.5 | 0.1 | 5.6 | 0.6 | 0.8 | 1.2 |
| LUH2V2H, Cropland, Net | 0.8 | 0.4 | 2.2 | 1.1 | 9.0 | 0.3 | 6.7 | 1.7 | 50.2 | 2.3 |
| LUH2V2H, Cropland, Gross | 0.8 | 0.4 | 2.2 | 1.1 | 9.0 | 0.3 | 6.7 | 1.7 | 50.2 | 2.3 |
| LUH2V2H, Grassland, Net | 1.3 | 0.1 | 0.7 | 0.9 | 0.8 | 0.2 | 0.1 | 3.0 | 78.4 | 0.5 |
| LUH2V2H, Grassland, Gross | 1.3 | 0.1 | 0.7 | 0.9 | 0.8 | 0.2 | 0.1 | 3.0 | 78.4 | 0.5 |
| Hansen, Forest, Net | 1.0 | 2.6 | 0.7 | 3.0 | 0.3 | 1.7 | 2.4 | 1.5 | 1.3 | 1.5 |
| Hansen, Forest, Gross | 1.6 | 3.8 | 1.7 | 4.0 | 0.9 | 2.5 | 3.7 | 2.5 | 1.8 | 2.4 |
| Houghton&Nassikas, Forest, Net | 1.1 | 0.04 | -0.4 | 0.3 | -0.5 | 0.04 | -0.3 | -0.1 | 0.7 | -0.1 |

---

## Editor Comment (EC1) · D. J. Carlson (Editor) · 11 Dec 2017

One reviewer asked to see the manuscript again after author response and changes. That reviewer has now conveyed this message:

"The authors answered all comments to my satisfaction and changed their text accordingly. I would like to recommend a publication of the manuscript ESSD-2017-74."

With that positive assessment now received and on the record, we will close discussion and move to final decision.

[Figure]

2017.